# Endocytic trafficking promotes vacuolar enlargements for fast cell expansion rates in plants

Kai Dünser[1,2,3], Maria Schöller[2], Ann-Kathrin Rößling[1,3], Christian Löfke[2], Nannan Xiao[2], Barbora Pařízková[4], Stanislav Melnik[2], Marta Rodriguez-Franco[5], Eva Stöger[2], Ondřej Novák[4], Jürgen Kleine-Vehn[1,2,3]*

[1]Molecular Plant Physiology (MoPP), Faculty of Biology, University of Freiburg, Freiburg, Germany; [2]Department of Applied Genetics and Cell Biology, University of Natural Resources and Life Sciences, Vienna, Austria; [3]Center for Integrative Biological Signalling Studies (CIBSS), University of Freiburg, Freiburg, Germany; [4]Laboratory of Growth Regulators, Institute of Experimental Botany of the Czech Academy of Sciences and Faculty of Science of Palacký University, Olomouc, Czech Republic; [5]Cell Biology, Faculty of Biology, University of Freiburg, Freiburg, Germany

*For correspondence: juergen.kleine-vehn@biologie. uni-freiburg.de

**Abstract** The vacuole has a space-filling function, allowing a particularly rapid plant cell expansion with very little increase in cytosolic content (Löfke et al., 2015; Scheuring et al., 2016; Dünser et al., 2019). Despite its importance for cell size determination in plants, very little is known about the mechanisms that define vacuolar size. Here, we show that the cellular and vacuolar size expansions are coordinated. By developing a pharmacological tool, we enabled the investigation of membrane delivery to the vacuole during cellular expansion. Our data reveal that endocytic membrane sorting from the plasma membrane to the vacuole is enhanced in the course of rapid root cell expansion. While this 'compromise' mechanism may theoretically at first decelerate cell surface enlargements, it fuels vacuolar expansion and, thereby, ensures the coordinated augmentation of vacuolar occupancy in dynamically expanding plant cells.

## Editor's evaluation

Plant cells can grow to extraordinarily large volumes; Arabidopsis root cells, for example, can expand beyond 50um long. Vacuole expansion is correlated with cell elongation, presumably to "fill up" the volume of the cell without requiring a tremendous volume of cytoplasm. Here, the authors carefully characterize an exciting new small molecule inhibitor of endocytic trafficking to the vacuole. This new tool will be valuable to researchers studying endocytic trafficking and vacuole biogenesis in plants.

## Introduction

Animals and plants take many shapes and come in great diversity of sizes (*Marshall et al., 2012*). During cellular expansion, the surface area to intracellular volume ratio gets smaller and if cells grow beyond a critical limit, the surface of the plasma membrane may cease to accommodate cellular needs. Cells typically remain, hence, relatively small or eventually induce membrane furcation to enlarge the cellular surface. Compared to animals, plant cells can dramatically increase their size without the apparent need for surface furcation. The vacuole, occupying up to 90% of a cellular volume, is the biggest organelle in plants and its size correlates with and determines the cell size (*Owens*

*and Poole, 1979*; *Berger et al., 1998*; *Löfke et al., 2015*; *Scheuring et al., 2016*). The size of the vacuole increases during cell expansion and vacuolar morphology is a reliable intracellular read-out for cellular expansion (*Dünser et al., 2019*). The dynamic regulation of vacuolar size allows plant cells to expand with little increase of the cytosol (*Dünser et al., 2019*), maintaining a favorable cell surface to cytosol ratio during growth. Despite its anticipated importance for fast cellular expansion, very little is known about how cells dynamically scale and coordinate its vacuolar size (*Dünser and Kleine-Vehn, 2015*; *Krüger and Schumacher, 2018*). Vacuolar soluble *N*-ethylmaleimide-sensitive-factor attachment receptors (SNAREs) reliant membrane remodeling as well as actin/myosin-dependent vacuolar constrictions define the auxin-dependent vacuolar occupancy of the cell (*Löfke et al., 2015*; *Scheuring et al., 2016*). The vacuolar SNARE complex, consisting of R-SNARE VAMP711, Qa-SNARE SYP22, Qb-SNARE VTI11, as well as Qc-SNAREs SYP51 and SYP52, regulates hetero- and homo-typic membrane fusion at the tonoplast (*Fujiwara et al., 2014*). Despite our molecular knowledge on vesicle trafficking toward the vacuole (*Krüger and Schumacher, 2018*; *Kang et al., 2022*), very little is known about the dynamics of membrane delivery to the vacuole and its implication for cellular expansion.

## Results

In this project, we hypothesized that the interference with vacuolar SNAREs could aid us to visualize the rate and importance of membrane delivery to the vacuole. Mutations in the vacuolar SNARE subunit *VTI11* leads to strongly affected, roundish, and partially fragmented vacuoles (*Yano et al., 2003*; *Zheng et al., 2014*; *Löfke et al., 2015*). Despite the strongly affected vacuolar morphology, the root length, as well as the size of fully elongated epidermal cells of untreated *vti11* mutants, were hardly distinguishable from wild-type seedlings (*Figure 1A-D*). This finding seemingly questions the contribution of vacuolar SNARE-dependent membrane fusions at the tonoplast for cellular expansion. Considering that the space-filling function of the vacuole defines cell size determination (*Dünser et al., 2019*), we next analyzed the contribution of *VTI11* for vacuolar occupancy of the cell, using confocal z-stack imaging and 3D renderings of epidermal atrichoblast cell files in the meristematic and elongation zone (*Figure 1E*). Although *vti11* mutant vacuoles are morphologically distinct from wild-type vacuoles (*Yano et al., 2003*; *Zheng et al., 2014*; *Löfke et al., 2015*), their vacuolar occupancy of epidermal root cells was not distinguishable from wild-type (*Figure 1F*). This finding suggests that redundancy may ensure the space-filling function of the *vti11* mutant vacuole and, thereby, likely safeguards cell expansion.

*VTI11* and *VTI12* are able to substitute each other in their respective SNARE complexes with the *vti11 vti12* double mutants being embryonic lethal (*Surpin et al., 2003*). In order to overcome genetic redundancy as well as lethality in the crucial control of vesicle trafficking toward the vacuole, we conducted a small molecule screen with the aim to identify compounds impacting on SNARE-dependent delivery of vesicles to the vacuole. For our primary screen, we germinated seedlings constitutively expressing the vacuolar R-SNARE marker pUBQ10::YFP-VAMP711 in growth medium supplemented with bioactive chemicals from a library of 360 compounds, which likely affect cell expansion in planta (*Drakakaki et al., 2011*; *Figure 2A*). We subsequently employed a fluorescence binocular to identify compounds that intensified the YFP-VAMP711 signal (*Figure 2B*). In a secondary, CLSM (confocal laser scanning microscopy)-based validation screen, we recorded vacuolar morphology changes in the samples treated with the compounds in comparison to solvent (control) treatments (*Figure 2C*, *Figure 2—figure supplement 1*). Thereby, we isolated 12 small molecules (*Figure 2—figure supplement 1*) that affected YFP-VAMP711 and vacuolar morphology (*Figure 2—figure supplement 1*, *Figure 2—figure supplement 2*), which we thus named vacuole affecting compounds (VACs) 1–12.

Here, we primarily report on VAC1 (*N*'-[(2,4-dibromo-5-hydroxyphenyl)methylidene]-2-phenylacetohydrazide), which affected vacuolar morphology (*Figure 2—figure supplement 1*) and induced ectopic accumulation of vacuolar SNARE marker YFP-VAMP711 in aster-like structures adjacent to the main vacuole (*Figure 2C*). Due to the possibly hydrolyzable acylhydrazone bond in VAC1, we wondered whether VAC1 could be potentially metabolized in planta into some of its sub-structures or other metabolic products. VAC1 can be synthesized from 2,4-dibromo-5-hydroxybenzaldehyde and phenylacetic acid (PAA) hydrazide, but the application of these substances (here referred to as precursors) did not cause any alterations to vacuolar morphology (*Figure 2—figure supplement 3A*). We also analyzed the potential VAC1 conversion to PAA, because it is a naturally occurring compound

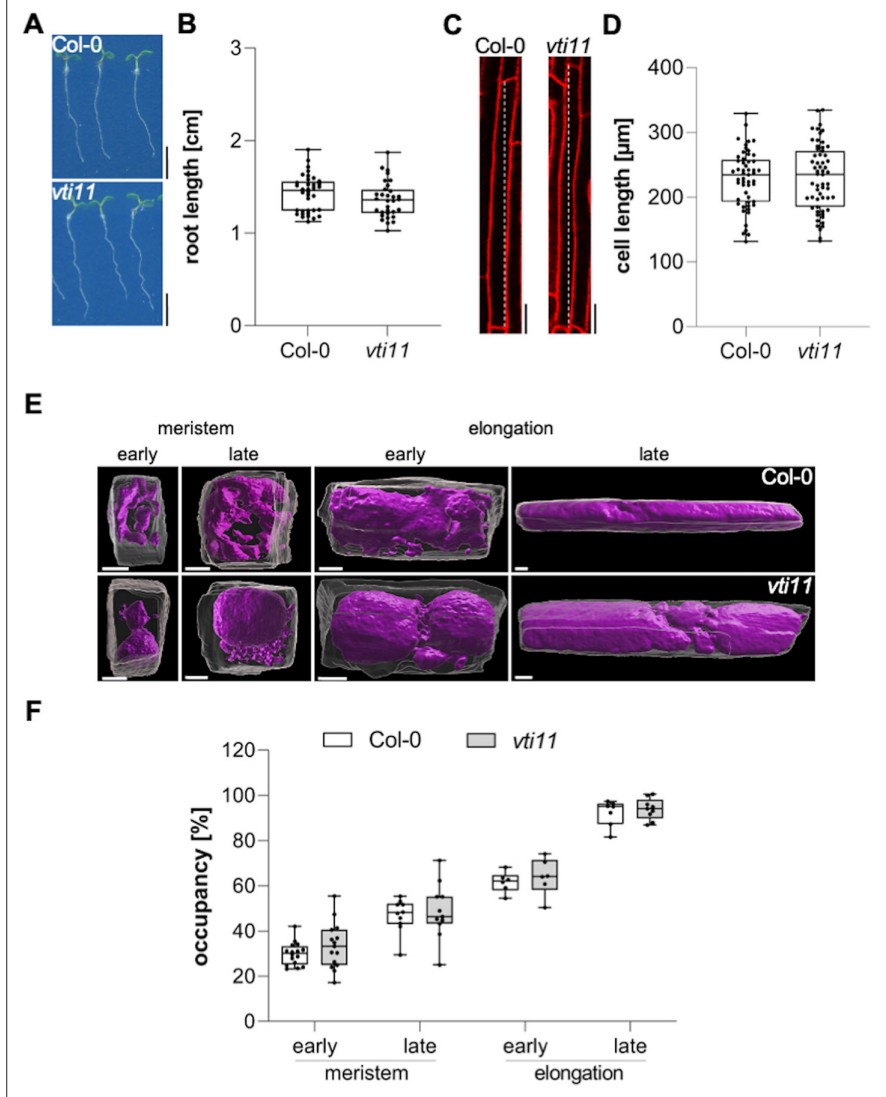

**Figure 1.** Vacuolar occupancy of the cell defines cell expansion. (**A**) Representative images (scale bars: 0.5 cm) and (**B**) quantification of the main root length of 6-day-old Col-0 (n=36) and *vti11* seedlings (n=32). Student's t-test (ns). (**C**) Representative images (scale bars: 25 µm) of differentiated atrichoblast cells. Seedlings were stained with propidium iodide (PI) (red) for 30 min in liquid medium prior image acquisition. (**D**) Boxplots show cell length quantification of 6-day-old Col-0 (n=53) and *vti11* (n=59) seedlings. Student's t-test (ns). (**E**) 3D reconstructions of PI-stained cell walls (gray) and BCECF-stained vacuoles (magenta) in the early and late meristem and in the early and late elongation zone. Scale bars: 5 µm. (**F**) Boxplots show vacuolar occupancy of cells in the defined zones (n=7–16). Student's t-test (ns). Boxplots: Box limits represent 25th percentile and 75th percentile; horizontal line represents median. Whiskers display min. to max. values. Data points are individual measured values. Representative experiments are shown.

The online version of this article includes the following source data for figure 1:

**Source data 1.** *Figure 1B* source data.

**Source data 2.** *Figure 1D* source data.

**Source data 3.** *Figure 1F* source data.

with low auxin activity (*Abe et al., 1974*). The ultra-high performance liquid chromatography-selected ion recording-mass spectrometry (UHPLC-SIR-MS) analysis however did not detect PAA in VAC1-treated seedlings (*Figure 2—figure supplement 3B-D*).

We conclude that VAC1 is not detectably converted to PAA under our conditions. On the other hand, in VAC1-treated samples, the UHPLC-UV-MS analysis revealed three additional peaks

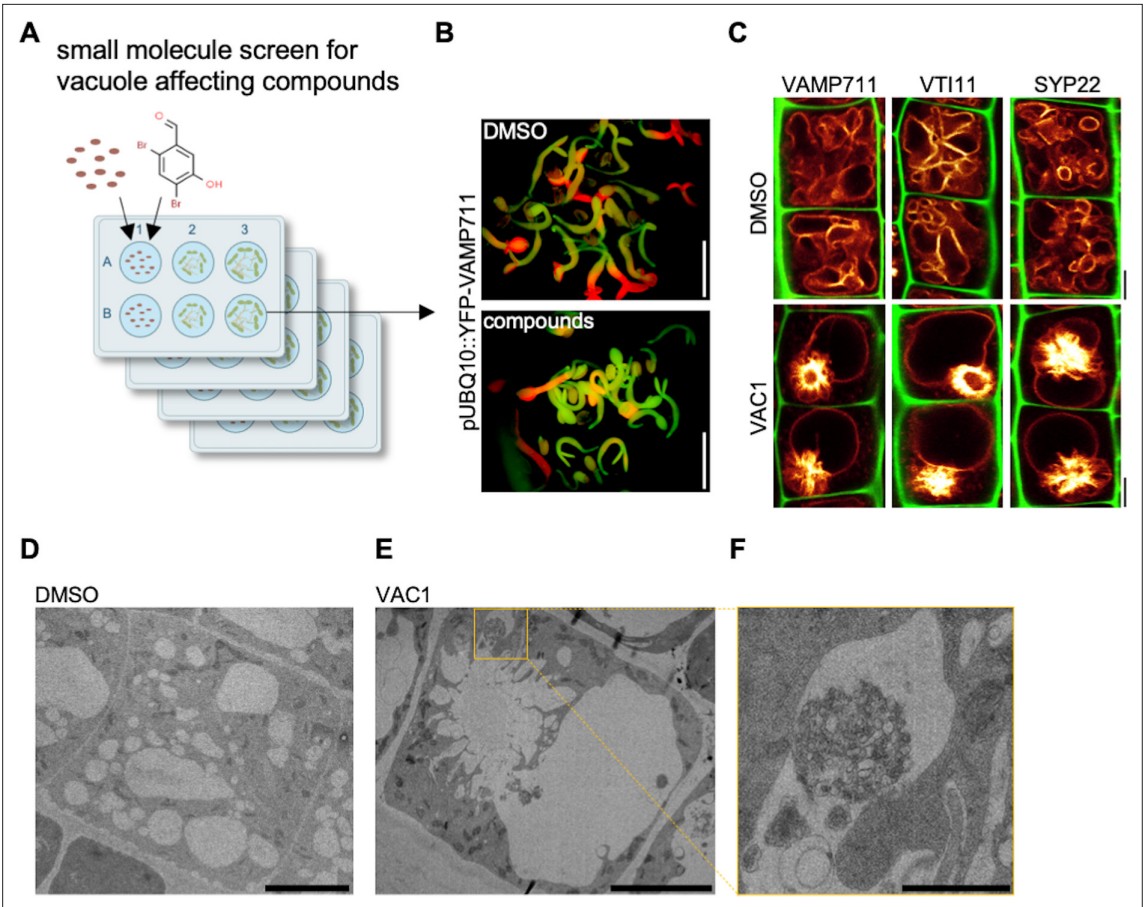

**Figure 2.** Bioactive small molecule screen identifies VAC1 as a vacuole affecting compound. (**A**) Schematic depiction of the small molecule screen workflow. pUBQ10::YFP-VAMP711 seeds were germinated in liquid medium containing solvent control dimethyl sulfoxide (DMSO) or 360 small molecules from a library of bioactive compounds. (**B**) Four-day-old seedlings were then screened for intensified YFP-VAMP711 signal using a fluorescence binocular. Scale bars: 0.5 cm. (**C**) Representative images of the confocal laser scanning microscopy (CLSM)-based confirmation screen. Cell wall and vacuolar membrane in late meristematic atrichoblast cells were visualized with propidium iodide (PI) (green) and pUBQ10::YFP-VAMP711 (yellow), pSYP22::SYP22-GFP (yellow), or pUBQ10::pHGFP-VTI11 (yellow), respectively. Scale bars: 5 µm. (**D–F**) Representative transmission electron microscopy (TEM) images of solvent control (DMSO)- (**D**) and VAC1-treated (**E**) epidermal root cells. (**F**) shows a magnified detail of (**E**). Scale bars: 5 µm (**D,E**) and 1 µm (**F**).

The online version of this article includes the following source data and figure supplement(s) for figure 2:

**Figure supplement 1.** Bioactive small molecules impact vacuolar morphology.

**Figure supplement 2.** Bioactive small molecules from the screen.

**Figure supplement 3.** Characterization of VAC1 stability in planta.

**Figure supplement 4.** Characterization of VAC1 derivatives (VAC1As).

**Figure supplement 4—source data 1.** *Figure 2—figure supplement 4C* source data.

**Figure supplement 5.** VAC1 induces ectopic accumulation of VHAa3-GFP.

for potential VAC1 metabolites (M1-M3) with a retention time of 4.37, 5.26, and 5.39 min for M1, M2, and M3, respectively (*Figure 2—figure supplement 3E-F*). High-resolution mass spectrometry (HRMS) identified these metabolites based on exact mass determination and characteristic distribution of bromine isotopes (*Figure 2—figure supplement 3G*). Metabolite M1 was predicted to be the hydroxylated form (VAC-OH, Rt 4.37 min, [M-H⁻]: 424.9129 Da, calculated elemental composition $C_{15}H_{11}Br_2N_2O_3$) and the other two derivatives relate to glycosylated forms. The detected compound M2 likely represented VAC1-3-*O*-glucoside (VAC1-Glc, Rt 5.26 min, [M-H⁻]: 570.9706 Da, $C_{21}H_21Br_2N_2O_7$) and M3 was identified as VAC1-3-O-6-*O*-acetylglucoside (VAC1-AcGlc, Rt 5.39 min, $C_{23}H_23Br_2N_2O_8$, [M-H⁻]: 612.9805 Da) (*Figure 2—figure supplement 3G*). Using UHPLC–UV ($\lambda_{max}$ = 291 nm for all

compounds), we revealed VAC1-AcGlc (56.3% ± 0.8%) as the most abundant metabolite, followed by VAC1-Glc (31.3% ± 0.4%) and VAC1 (12.4% ± 0.6%), while the UV signal of hydroxylated VAC1-OH (<0.1%) was close to the detection limit of the photodiode array detector used (*Figure 2—figure supplement 3H*). This set of data shows that a substantial amount of VAC1 undergoes glycosylation in planta, which could affect its activity in long-term experiments.

Next, we used structurally related derivatives of VAC1, termed VAC1-analogue (VAC1A), to further address whether the entity of VAC1 is required for its effects on vacuolar morphology (*Figure 2—figure supplement 4A*). At the concentration and treatment time used for the subcellular effect of VAC1 (10 µM for 2.5 hr), solely VAC1A4, which displays an additional hydroxy group, induced severe vacuolar morphology defects (*Figure 2—figure supplement 4B*). The cellular effects of VAC1A4 were, however, visually distinct from VAC1. While short-term (2.5 hr) VAC1A4-treated cells appeared viable (based on the intact extracellular propidium iodide [PI] staining) (*Figure 2—figure supplement 4B*), the main root growth did not resume after wash-out (*Figure 2—figure supplement 4C*), indicating some degree of toxicity. On the other hand, interference with either the phenyl group or the functional groups of the 2,4-dibromo-5-hydroxybenzaldehyde reduced the in planta activity and/or stability of VAC1 (*Figure 2—figure supplement 4B*). At higher concentrations (25 and 50 µM), VAC1A2 and VAC1A5 also impacted to some degree on vacuolar morphology, while VAC1A1 and VAC1A3 appeared largely inactive in regard to vacuolar morphology (*Figure 2—figure supplement 4B*). These results imply that the entity of VAC1 (phenyl- as well as the benzaldehyde rings) contributes to its effect on vacuolar morphology.

Next, we addressed the subcellular effects of VAC1 on the endomembrane system. When compared to YFP-VAMP711, we observed similar VAC1 effects on vacuolar morphology and strong aster-like protein accumulation of *pUBQ10::pHGFP-VTI11-* and *pSYP22::SYP22-GFP-* (*Figure 2C*). This finding suggests that VAC1 affects the distribution of vacuolar SNARE complex components. VAC1 also induced similar ectopic accumulation of vacuolar membrane marker VHA-a3-GFP (*Figure 2—figure supplement 5*), suggesting that other tonoplast proteins are not excluded from the region of aster-like aggregations and that these structures are part of the tonoplast. To obtain a better understanding of the VAC1-reliant subcellular defects, we subsequently performed electron microscopy. When compared to control treatments, VAC1 induced ectopic extensions of the vacuole, which were reminiscent to the before mentioned aster-like structures (*Figure 2D and E*). In addition, we observed VAC1-dependent clusters of vesicles in close proximity to the malformed vacuole (*Figure 2F*), possibly visualizing defective vesicle delivery to the vacuole. To address whether VAC1 disrupts vesicle trafficking to the vacuole, we next employed the styryl dye FM4-64 (*Vida and Emr, 1995*; *Scheuring et al., 2015*). The endocytic tracer FM4-64 initially labels the plasma membrane and is rapidly internalized following the endocytic pathway, eventually staining the vacuolar membrane (*Ueda et al., 2001*; *Scheuring et al., 2015*). Tonoplast labelling with FM4-64 was observed within 3 hr following the pulse staining in mock-treated seedlings (*Figure 3A*). Strikingly, staining of the vacuolar membrane was absent in VAC1-treated samples and instead FM4-64 accumulated in roundish structures (VAC1 bodies) (*Figure 3A*). This set of data demonstrates that VAC1 treatments prevent the fusion of FM4-64-positive endocytic vesicles to the central vacuole.

Next, we inspected plasma membrane resident proteins BRI1-GFP, PGP19-GFP, and NPSN12-GFP, expecting a fraction of these proteins transiting to the vacuole for lytic degradation (*Kleine-Vehn et al., 2008*). When compared to the ectopic FM4-64 accumulations, the examined lines showed comparably weak, but clearly detectable VAC1-induced aggregation of the fluorescently tagged proteins, which were absent in the solvent control treatments (*Figure 3B*). This finding corroborates our assumption that VAC1 interferes with vesicle trafficking to the vacuole. On the other hand, VAC1 did not visibly affect clathrin light chain marker CLC-GFP and F-actin marker Lifeact-venus (*Figure 3C*). Similarly, early and late endosome marker lines, such as GOT1-YFP (WAVE18Y), VHA-a1-GFP, and RABF2A-YFP (RHA1), were not disrupted upon VAC1 administration (*Figure 3D*). This set of data suggests that early vesicle sorting toward the vacuole is likely not disturbed, but stalls in close proximity to the vacuole.

To further assess VAC1-sensitive vesicle trafficking, we initially used brefeldin A (BFA), which is a specific inhibitor of ARF-GEFs (ADP-ribosylation factor – guanine-nucleotide exchange factors) and blocks endocytic trafficking toward the vacuole at the level of the trans-Golgi network (TGN) (*Kleine-Vehn et al., 2008*). The VAC1-induced accumulation of vacuolar SNARE SYP21-GFP surrounded the

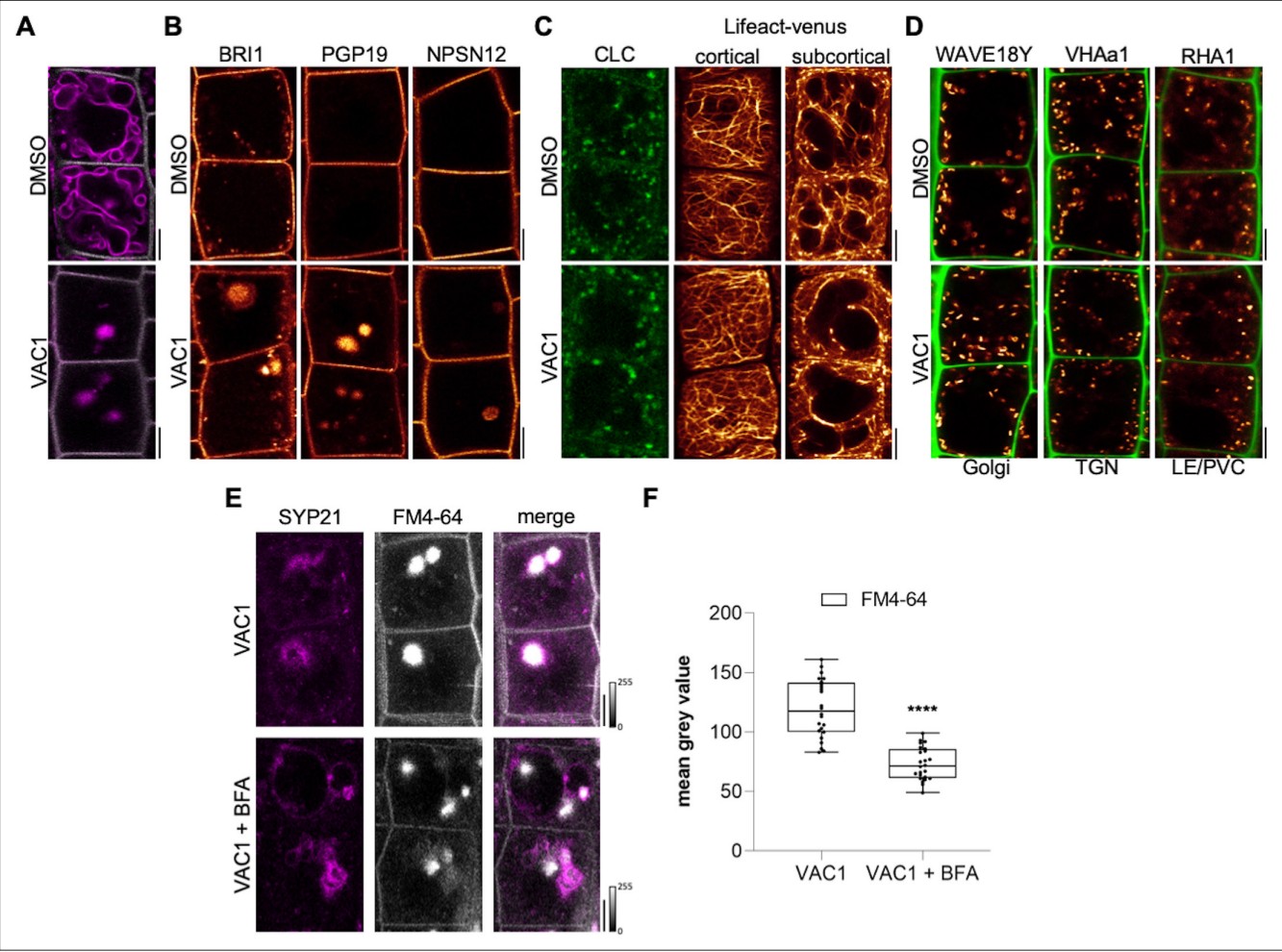

**Figure 3.** VAC1 interferes with vesicle trafficking to the vacuole. (**A**) Tonoplast and plasma membrane of late meristematic atrichoblast cells were visualized with FM4-64 (magenta) and pUBQ10::NPSN12-YFP (gray), respectively. Six-day-old seedlings were pre-treated with dimethyl sulfoxide (DMSO) or 10 µM VAC1 for 30 min., then pulse-stained with 4 µM FM4-64 for 5 min and subsequently transferred to liquid medium containing DMSO or 10 µM VAC1 for 3 hr. Scale bars: 5 µm. (**B**) Representative images of 6-day-old seedlings of plasma membrane marker lines pBRI1::BRI1-GFP (yellow), pPGP19::PGP-19-GFP (yellow), and pUBQ10::NPSN12-YFP (yellow). Seedlings were treated with DMSO or 10 µM VAC1 for 2.5 hr in liquid medium. Scale bars: 5 µm. (**C**) Representative images of 6-day-old seedlings of CLC-GFP (green) and Lifeact-venus (yellow) marker lines. Seedlings were treated with DMSO or 10 µM VAC1 for 2.5 hr in liquid medium. Scale bars: 5 µm. (**D**) Representative images of 6-day-old WAVE18Y (pUBQ10::GOT1-YFP) (yellow), pVHAa1::VHAa1-GFP (yellow), and WAVE7Y (pUBQ10::RABF2a-YFP) (yellow) treated with DMSO or 10 µM VAC1 for 2.5 hr in liquid medium. Cell walls were counterstained with propidium iodide (PI). Scale bars: 5 µm. (**E**) Six-day-old p35S::SYP21-GFP (magenta) seedlings were pulse-stained with 4 µM FM4-64 (gray) for 5 min and subsequently pre-treated with 25 µM BFA for 30 min before adding 10 µM VAC1 or solvent control for another 60 min. The treatments were conducted in liquid medium. Scale bars: 5 µm. (**F**) Boxplots show mean gray values of FM4-64 fluorescence signal in VAC1 bodies in VAC1 (n=24) or VAC1 + BFA (n=24) treatments. Student's t-test (****p<0.0001). Boxplots: Box limits represent 25th percentile and 75th percentile; horizontal line represents median. Whiskers display min. to max. values. Data points are individual measured values. Boxplots: Box limits represent 25th percentile and 75th percentile; horizontal line represents median. Whiskers display min. to max. values. Data points are individual measured values.

The online version of this article includes the following source data for figure 3:

**Source data 1.** *Figure 3F* source data.

FM4-64 accumulation in VAC1 bodies, which was significantly reduced in BFA co-treatments (*Figure 3E and F*). We, hence, conclude that VAC1 affects vacuolar vesicle trafficking downstream of the TGN. To further decipher the VAC1 effect downstream of the TGN, we next addressed its dependency on the CLASS C CORE VACUOLE/ENDOSOME TETHERING (CORVET) and homotypic fusion and protein sorting (HOPS) machineries (*Takemoto et al., 2018*), because these membrane tethering complexes supply the vacuolar SNAREs with heterotypic (endosomal derived) and homotypic (vacuole derived) membranes, respectively. The dexamethasone (DEX)-inducible amiRNA lines that target either the

CORVET-specific subunit VPS3 or the HOPS-specific subunit VPS39 enable the selective repression of the CORVET and HOPS complex, respectively (*Takemoto et al., 2018*). The transcriptional repression of CORVET or HOPS inhibits root growth (*Figure 4A and B* and *Figure 4—figure supplement 1A-C*). In agreement, seedlings germinated on VAC1 containing media also displayed reduced main root growth (*Figure 4A, B, F, G*), suggesting that vesicle trafficking to the vacuole contributes to root growth. Homotypic membrane fusion unlikely contributes to the VAC1 effect, because VPS39 depleted roots remained fully sensitive to VAC1 (*Figure 4—figure supplement 1A-C*). In contrast, the repression of CORVET function induced a partially resistant root growth to VAC1 (*Figure 4A–C*), suggesting that defects in heterotypic vesicle trafficking toward the vacuole contribute to the VAC1-dependent interference with root growth. To visualize vacuolar morphology after CORVET repression, we used the pUBQ10::YFP-VAMP711 marker crossed to the DEX-inducible VPS3 amiRNA line (*Takemoto et al., 2018*). The conditional knockdown of VPS3 had no visible effect on the distribution of *YFP-VAMP711* and the VPS3-deprived cells remained sensitive to VAC1-induced aberrations in VAMP711 localization (*Figure 4D*).

We, accordingly, propose that VAC1 primarily interferes with vesicle fusion at the tonoplast downstream of heterotypic vesicle tethering complex. In agreement, VAC1-treated wild-type cells remarkably resembled the vacuolar morphology of untreated *vti11* mutants (*Figure 4E*) and, hence, we subsequently addressed the contribution of vacuolar SNARE *VTI11* to the VAC1 effect. We concluded that VAC1 attenuates root growth rates in a VTI11-dependent manner, because when compared to wild-type seedlings the *vti11* mutants were less sensitive to VAC1 (*Figure 4E–H*). On the other hand, VAC1 application still induced alterations in VAMP711 localization (*Figure 4E*) and also root growth remained at least partially inhibited (*Figure 4F–H*) in the *vti11* mutant background. We accordingly conclude that VAC1 interferes with the redundant, SNARE-dependent membrane delivery to the vacuole. Accordingly, we propose that VTI11 is not the sole or main target of VAC1, assuming functional redundancy at the level of vacuolar SNAREs and/or other accessory molecular targets.

Considering that VAC1 interferes with vacuolar SNARE-dependent vesicle trafficking to the vacuole, this pharmacological tool subsequently allowed us to address the importance of vesicle trafficking toward the vacuole in elongating cells. We next utilized VAC1 to visualize the rate of endocytic trafficking to the vacuole in early and late meristematic as well as early and late elongation zones. We therefore used plasma membrane marker BRI1-GFP and FM4-64-based pulse labelling of plasma membranes (*Figure 5—figure supplement 1A-D*; *Figure 5A–I*). Elongating cells displayed higher BRI1-GFP and FM4-64 uptake into VAC1 bodies when compared to meristematic cells (*Figure 5—figure supplement 1C, D* and *Figure 5A and B*). This finding implies either meristematic and elongating cells to differ in endocytic membrane trafficking or alternatively inhomogeneous VAC1 activity/uptake. To further address the possibility of altered endomembrane sorting during cellular elongation, we similarly used the well characterized inhibitor BFA to block membrane recycling and sorting toward the vacuole at the level of the TGN (*Kleine-Vehn et al., 2008*). In line with the results from VAC1 treatments, BFA bodies also showed greater FM4-64 signal accumulation in cells of the elongation zone when compared to cells of the meristem (*Figure 5C and D*). We, hence, propose that BFA and VAC1 visualize enhanced membrane sorting from the plasma membrane toward the vacuole during cellular elongation.

Due to the steady surface increase, elongating cells may intrinsically show enhanced vesicle trafficking to the vacuole, even if endocytic membrane delivery would remain constant. Hence, we next performed whole-cell 3D reconstructions of early as well as late meristematic cells and observed that these cells displayed roughly a doubling of surface area and volume (*Figure 5E and F*). Compared to the cellular size increase, the volume of FM4-64 containing BFA bodies showed a much stronger relative size increase (*Figure 5E and F*). Considering that the volume of the BFA compartment is filled with small vesicles (*Geldner et al., 2001*), we anticipate that cellular surface enlargement alone does not fully account for the observed alterations in intracellular dynamics. We, hence, rather conclude that endocytic membrane sorting toward the vacuole undergoes reprogramming in elongating cells.

Next, we tested whether membrane delivery from the plasma membrane to the vacuole is enhanced during cellular elongation when we do not apply vesicle trafficking inhibitors. To this end, we quantified the co-localization of FM4-64 with the tonoplast marker YFP-VAMP711 in meristematic and elongating cells (*Figure 5G*). Three hours after the relatively equal pulse staining of FM4-64 (*Figure 5—figure supplement 1A, B*), the co-localization between FM4-64 and YFP-VAMP711 at

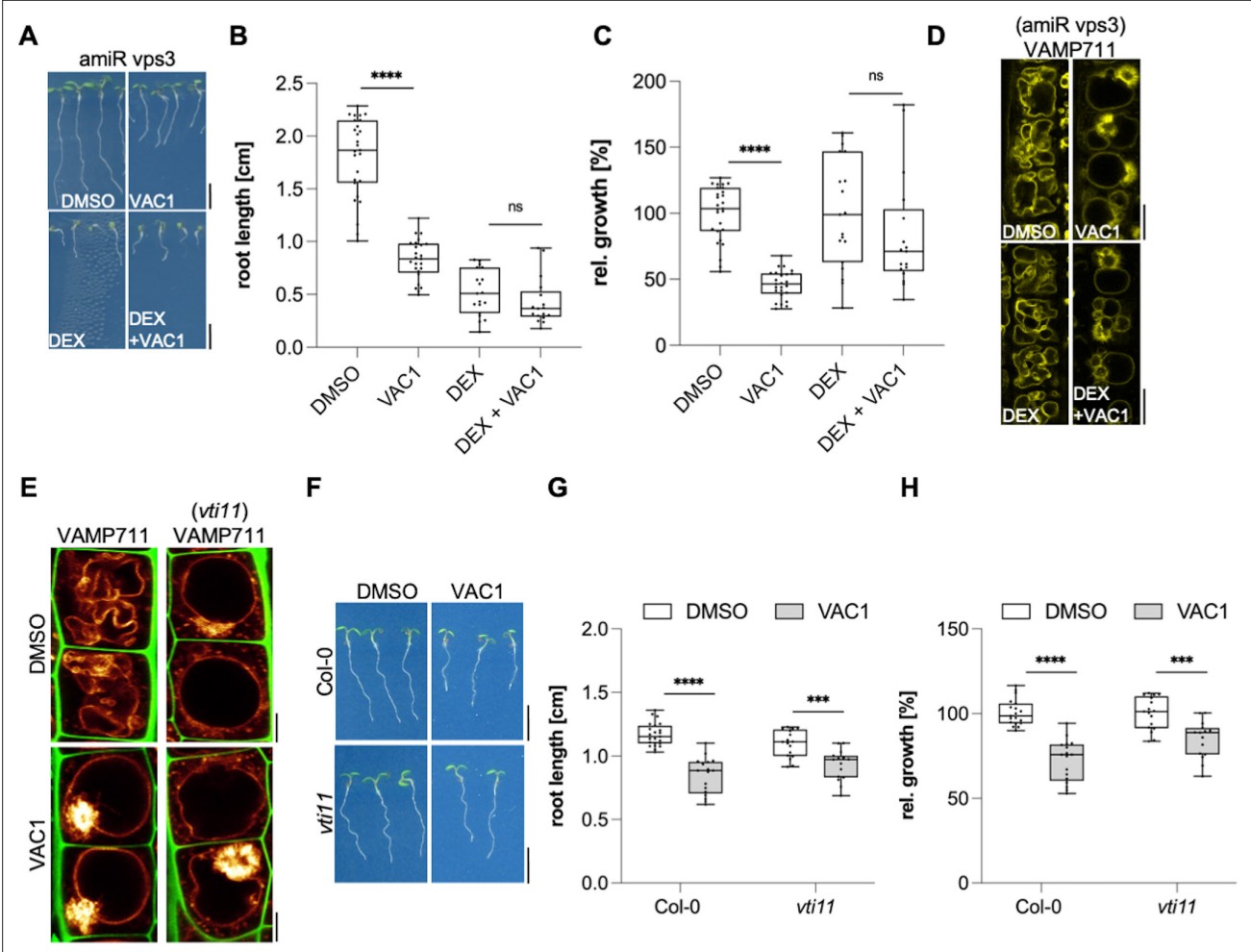

**Figure 4.** VAC1 specifically interferes with vacuolar soluble *N*-ethylmaleimide-sensitive-factor attachment receptor (SNARE)-dependent vesicle fusion to the tonoplast. (**A**) Representative images (scale bars: 0.5 cm) and (**B**) quantification of main root length of 7-day-old amiR Vps3 seedlings germinated on solvent control medium (dimethyl sulfoxide [DMSO], n=27), 10 µM VAC1 (n=25), 30 µM dexamethasone (DEX, n=19) or 10 µM VAC1 and 30 µM DEX (n=17). Two-way ANOVA with Sidaks's multiple comparisons test (DMSO to VAC1: ****p<0.0001, DEX to DEX + VAC1: p=0.3247). (**C**) Boxplots show relative growth of VAC1 samples and DEX + VAC1 samples. Statistics is based on two-way ANOVA and multiple comparison analysis (DMSO to VAC1: ****p<0.0001, DEX to DEX + VAC1: p=0.0660). (**D**) Representative images of late meristematic atrichoblast cells in *amiR Vps3 x pUBQ10::YFP-VAMP711* seedlings. Seedlings were grown for 4 days, then transferred to plates containing solvent control (DMSO) or 30 µM DEX for another 3 days and subsequently treated for 2.5 hr with DMSO, 30 µM DEX, 10 µM VAC1 or VAC1 + DEX, respectively, in liquid medium. Scale bars: 5 µm. (**E**) Representative images of late meristematic atrichoblast cells of pUBQ10::YFP-VAMP711 in wild-type and *vti11* background. Propidium iodide (PI) (green) and *pUBQ10::YFP-VAMP711* (yellow) depict cell wall and vacuolar membrane, respectively. Six-day-old seedlings were treated with solvent control (DMSO) or 10 µM VAC1 for 2.5 hr in liquid medium. Scale bars: 5 µm. (**F**) Representative images (scale bars: 0.5 cm) and (**G**) boxplots showing main root length of 6-day-old Col-0 and *vti11* seedlings grown on solvent control (DMSO) (n=23 for Col-0, n=17 for *vti11*) or 20 µM VAC1 (n=17 for Col-0, n=15 for *vti11*) plates. Statistical significance was determined by two-way ANOVA with Sidaks's multiple comparisons test (Col-0: ****p<0.0001, *vti11*: ***p=0.0006). (**H**) Boxplots show relative growth of Col-0 and *vti11* on 20 µM VAC1. Statistics is based on two-way ANOVA and multiple comparison analysis (Col-0: ****p<0.0001, *vti11*: ***p=0.0001). Boxplots: Box limits represent 25th percentile and 75th percentile; horizontal line represents median. Whiskers display min. to max. values. Data points are individual measured values. Boxplots: Box limits represent 25th percentile and 75th percentile; horizontal line represents median. Whiskers display min. to max. values. Data points are individual measured values.

The online version of this article includes the following source data and figure supplement(s) for figure 4:

**Source data 1.** *Figure 4B, C* source data.

**Source data 2.** *Figure 4G,H* source data.

**Figure supplement 1.** VPS39-deprived roots are sensitive to VAC1.

**Figure supplement 1—source data 1.** *Figure 4—figure supplement 1B, C* source data.

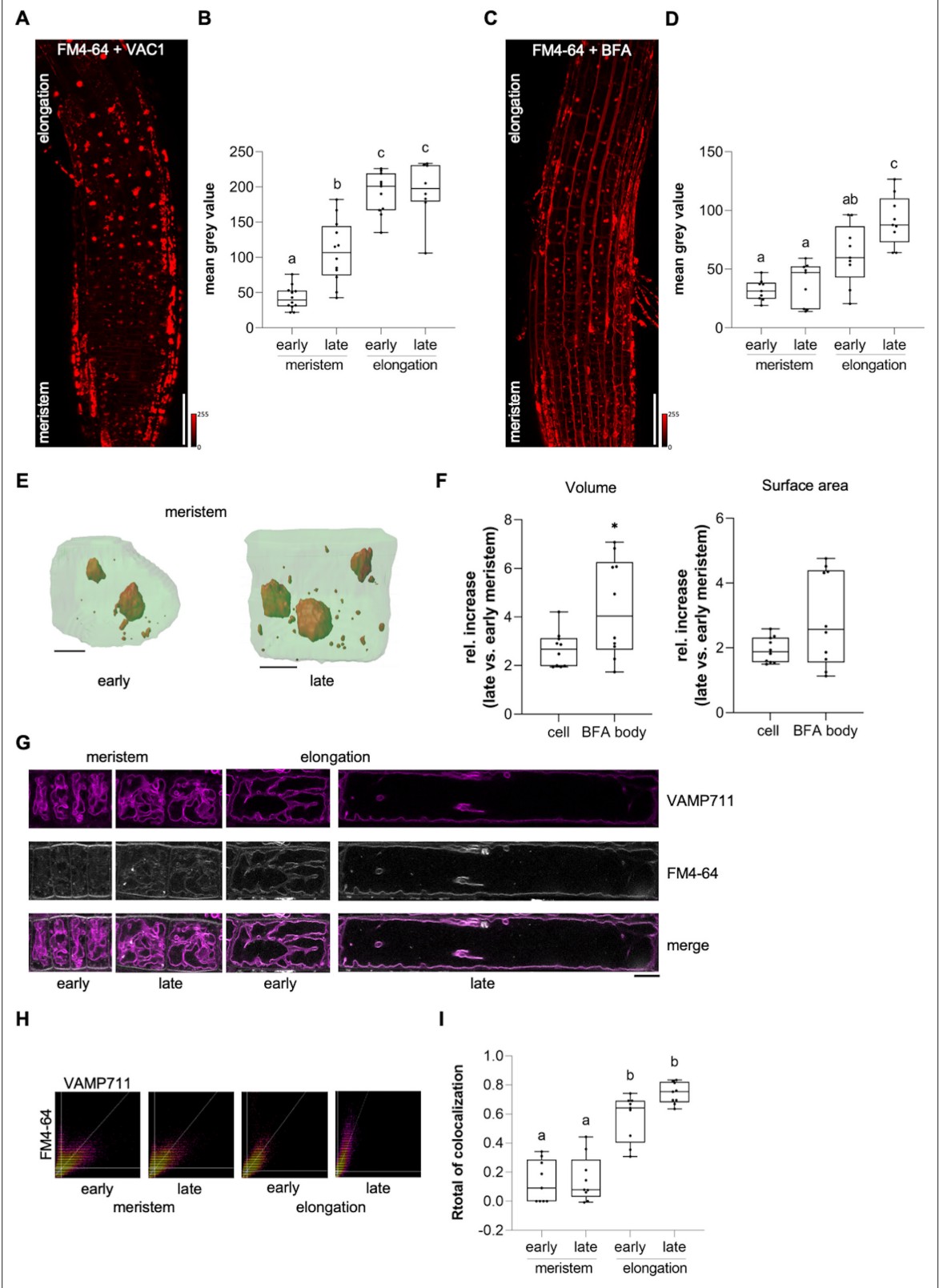

**Figure 5.** Endocytic trafficking is enhanced at the onset of cellular elongation. (**A**) Representative maximum z-projection (scale bar: 50 μm) of 6-day-old Col-0 roots. Seedlings were pre-treated with 10 μM VAC1 for 30 min before staining in 4 μM FM4-64 (red) for 30 min. Seedlings were then de-stained for 1.5 hr and subsequently treated with 10 μM VAC1 for 3 hr before image acquisition. The treatments and the staining were conducted in liquid medium. (**B**) Boxplots showing mean gray values of FM4-64 fluorescence signal in VAC1 bodies in early (n=12) and late (n=12) meristematic

*Figure 5 continued on next page*

*Figure 5 continued*

cells as well as in the early (n=11) and late (n=8) elongation zone. One-way ANOVA with Tukey's multiple comparisons test (b: p=0.0002, c: p<0.0001). (**C**) Representative maximum z-projection (scale bar: 50 μm) of 6-day-old Col-0 roots. Seedlings were pulse-stained in 4 μM FM4-64 (red) for 5 min and subsequently transferred to 25 μM BFA for 30 min prior to image acquisition. Staining and treatment was done in liquid medium. (**D**) Boxplots showing mean gray values of FM4-64 fluorescence signal in BFA bodies in early (n=9) and late (n=9) meristematic cells as well as in the early (n=9) and late (n=9) elongation zone. One-way ANOVA with Tukey's multiple comparisons test (b to a [early meristem]: p=0.0123, c to a: p<0.0001, c to b: p=0.0178). (**E**) 3D reconstructions of plasma membrane (pUBQ10::NPSN12-YFP, green) and BFA bodies (red) in early (left) and late (right) meristematic cells. Six-day-old pUBQ10::NPSN12-YFP seedlings were pulse-stained with 4 μM FM4-64 for 5 min and subsequently transferred to 50 μM BFA for 2 hr prior image acquisition. Staining and treatment was done in liquid medium. Scale bars: 5 μm. (**F**) Boxplots depict relative increase of cell and BFA body volume (n=10, respectively) and relative increase of cell and BFA body surface area (n=10, respectively) in late meristematic cells when compared to early meristematic cells. Student's t-test (*p=0.0217). (**G**) Representative images of early and late meristematic cells and cells of the early and late elongation zone. Six-day-old pUBQ10::YFP-VAMP711 seedlings were pulse-stained with 4 μM FM4-64 for 5 min and subsequently de-stained in liquid medium for 3 hr before image acquisition. Overlay of YFP-VAMP711 (magenta, upper panel) and FM4-64 (gray, middle panel) fluorescence signals is shown in the lower panel. Scale bar: 10 μm. Note: Images are assembled on white background. (**H**) Representative scatter plots depict co-localization of YFP-VAMP711 and FM4-64 in cells of the early and late meristem and in the early and late elongation zone. (**I**) Boxplots show Rtotal of co-localization between YFP-VAMP711 and FM4-64 in early (n=9) and late (n=9) meristematic cells and in the early (n=9) and late (n=9) elongation zone. One-way ANOVA with Tukey's multiple comparisons test (b: p<0.0001). Boxplots: Box limits represent 25th percentile and 75th percentile; horizontal line represents median. Whiskers display min. to max. values. Data points are individual measured values. Boxplots: Box limits represent 25th percentile and 75th percentile; horizontal line represents median. Whiskers display min. to max. values. Data points are individual measured values.

The online version of this article includes the following source data and figure supplement(s) for figure 5:

**Source data 1.** *Figure 5B* source data.

**Source data 2.** *Figure 5D* source data.

**Source data 3.** *Figure 5F* source data.

**Source data 4.** *Figure 5I* source data.

**Figure supplement 1.** FM4-64 evenly stains transversal plasma membranes in the meristem and elongation zone.

**Figure supplement 1—source data 1.** *Figure 5—figure supplement 1B* source data.

**Figure supplement 1—source data 2.** *Figure 5—figure supplement 1D* source data.

the vacuolar membrane was significantly lower in the meristematic zone when compared to the elongation zone (*Figure 5H, I*). Taken together, these results indicate that endocytic membrane sorting from the plasma membrane toward the vacuole undergoes significant rate changes during cellular elongation.

Based on our data, it is conceivable that endocytic trafficking impacts cell expansion via its contribution to vacuolar size. To approach this, we initially quantified cell and vacuole surface areas of (early and late) meristematic and (early and late) elongating cells using 3D reconstructions. We found that the area increase of both cell and vacuole surface is remarkably similar during the course of cell expansion (*Figure 6A* left panel, B). To address this coordinative mechanism, we applied VAC1 and employed whole-cell 3D reconstructions (*Figure 6A* right panel). Within only 2.5 hr, VAC1 induced an already detectable imbalance of cell and tonoplast surface size (*Figure 6C*). This result indicates that intensified membrane delivery is not only required at the plasma membrane, but similarly occurs at the tonoplast during cellular expansion. Accordingly, not only actin- and myosin-dependent vacuolar unfolding (*Scheuring et al., 2016*), but also membrane delivery greatly contributes to vacuolar size control.

Our data propose that the dynamic regulation of membrane sorting from the plasma membrane toward the vacuole is substantially contributing to the vacuole size increase, which could have an impact on rapid cell expansion rates. To address this, we used VAC1 to temporally block the vesicle fusion to the tonoplast during cell expansion. Accordingly, we mounted Col-0 seedlings in VAC1 or solvent (control) containing slide chambers and immediately inspected root growth under these conditions using confocal microscopy. VAC1 induced without a further delay a slower root growth (tip displacement) (*Figure 6D and E*), which correlated with reduced cellular expansion rates (*Figure 6F and G*) when compared to the solvent control. Accordingly, we conclude that vesicle trafficking to the vacuole is required for fast cellular expansion rates. In summary, our set of data suggests that endocytic trafficking toward the vacuole coordinates cellular and vacuolar surface increase, enabling rapid cell enlargements in plants.

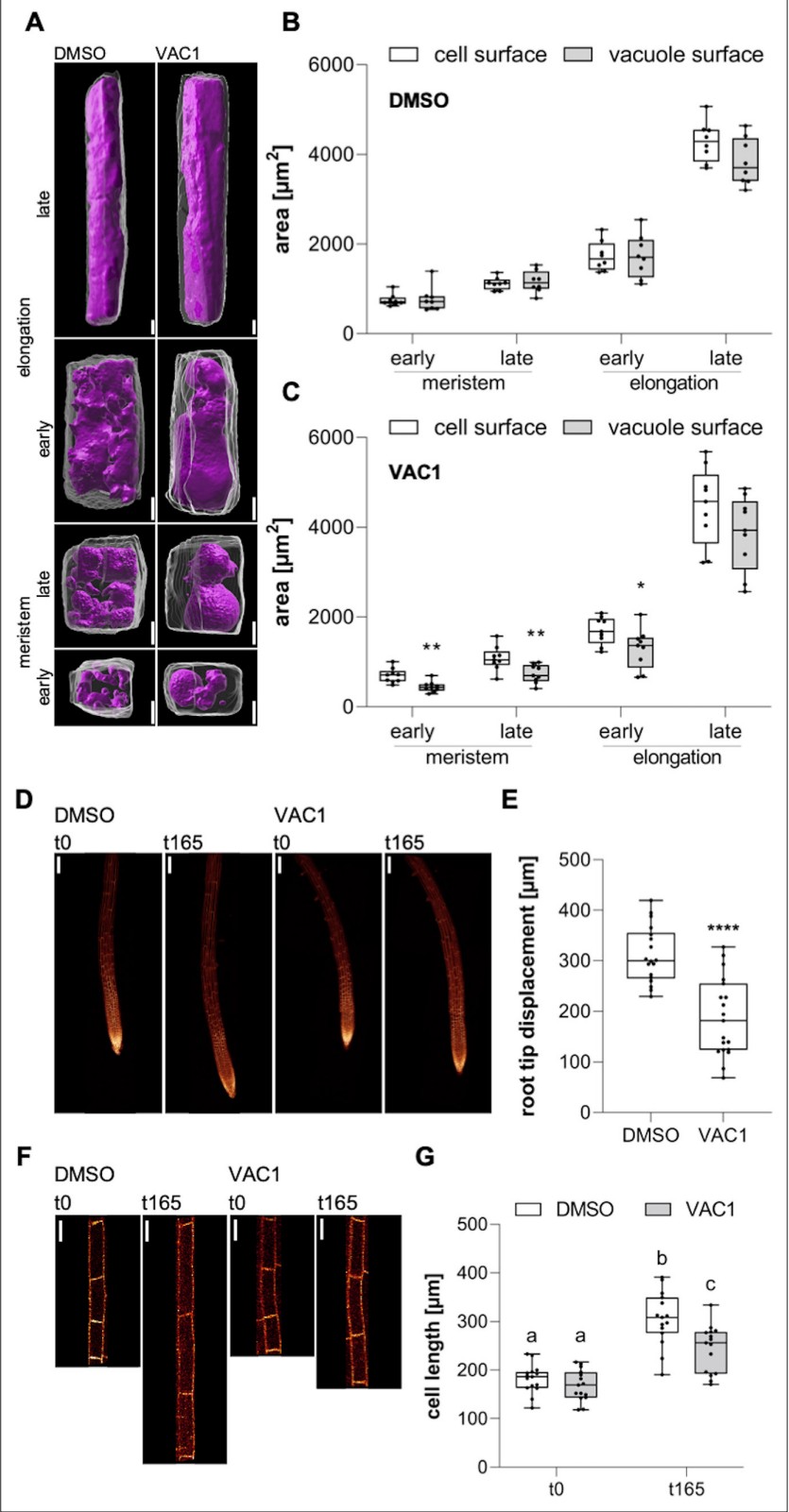

**Figure 6.** Coordinated surface increase at plasma membrane and tonoplast during cell elongation. (**A**) 3D reconstructions of propidium iodide (PI)-stained cell walls (gray) and BCECF-stained vacuoles (magenta) in the early and late meristem and in the early and late elongation zone. Scale bars: 5 µm. (**B**) and (**C**) Boxplots show cell and vacuole surface areas in the defined zones and indicated treatments. Six-day-old Col-0 seedlings were treated

*Figure 6 continued on next page*

*Figure 6 continued*

with solvent control (dimethyl sulfoxide [DMSO], n=8) or 10 µM VAC1 (n=9) for 2.5 hr in liquid medium. Student's t-test (early meristem: **p=0.0018, late meristem: **p=0.0073, early elongation: *p=0.0423). (**D**) Representative images (scale bars: 100 µm) of maximum z-projections of 6-day-old pUBQ10::NPSN12-YFP (yellow) seedlings. Seedlings were mounted on agar blocks containing solvent control (DMSO) or 20 µM VAC1 in chamber slides. t0 represents the first and t165 the last image acquisition timepoint, given in minutes after mounting. Note: Images are assembled on black background. (**E**) Boxplots show root tip displacement of DMSO (n=18) and VAC1 (n=19) treated samples at the end of the 165 min time frame. Student's t-test (****p<0.0001). (**F**) Representative images (scale bars: 25 µm) of maximum z-projections of 6-day-old pUBQ10::NPSN12-YFP (yellow) seedlings. Seedlings were mounted on agar blocks containing solvent control (DMSO) or 20 µM VAC1 in chamber slides. t0 represents the first and t165 the last image acquisition timepoint, given in minutes after mounting. Note: Images are assembled on black background. (**G**) Boxplots show cell lengths of DMSO (n=15) and VAC1 (n=15) treated samples at the beginning and the end of the 165 min time frame. One-way ANOVA with Tukey's multiple comparisons test (b to a: p<0.0001, b to c: p=0.0013, c to a: p≤0.0012).

The online version of this article includes the following source data for figure 6:

**Source data 1.** *Figure 6B, C* source data.

**Source data 2.** *Figure 6E* source data.

**Source data 3.** *Figure 6G* source data.

## Discussion

Here, we isolated and characterized VAC1, which interferes with the final steps of membrane delivery to the tonoplast. The molecular target of VAC1 remains currently unknown, but the structure-activity assessment suggests that the entirety of the compound is contributing to its activity, possibly insulating a complex mode of VAC1 action. Our data suggest that VAC1 disrupts SNARE-dependent, heterotypic vesicle delivery to the tonoplast. It is tempting to speculate that VAC1 interferes with possibly several subunits of the vacuolar SNARE complex or yet to be identified accessory proteins. Accordingly, in-depth characterization of the mode of VAC1 action could reveal intermolecular mechanisms of vesicle fusion at the tonoplast.

We used VAC1 to visualize the rate of vesicle trafficking toward the vacuole. Mechanistically, our data suggests that endocytic trafficking is accelerated at the onset of elongation, suggesting that plasma membrane-derived vesicles at least in part fuel the size increase of the vacuole. We propose that cell and vacuolar surface enlargements are thereby coordinated and pinpoint the importance of vacuolar size control for cell expansion and overall root growth. It remains to be seen how plant cells molecularly reprogram and monitor the rate of endocytic trafficking for ensuring fast cellular elongation. Not only VAC1 but also BFA treatments visualized higher rates of endocytic membrane accumulation in expanding cells. We, therefore, assume that the proposed reprogramming of membrane sorting occurs upstream or at the level of the TGN. The accelerated endocytic membrane flow toward the vacuole could relate to rate changes in endocytosis, but alternatively also the endosomal sorting machinery could shift the balance between endocytic recycling (back to the plasma membrane) and membrane drainage toward the vacuole.

The enhanced delivery of plasma membrane-derived membranes toward the vacuole during cell expansion seems somewhat counterintuitive, because it appears to slow down the theoretically possible cell surface increase. In addition, a high degree of membrane recycling at the plasma membrane is needed to fuel the high demand for cell wall synthesis (*Robert et al., 2005*; *Ketelaar et al., 2008*; *Ebine and Ueda, 2015*) and, hence, draining membranes toward the vacuole may even constrain the crucial cell wall remodeling during cell expansion. On the other hand, this vesicle trafficking-based compromise model for enlargements of plant cells allows the steady increase in vacuolar occupancy in dynamically expanding cells, which enables growth with little de novo synthesis of cytosolic components. This coordinated membrane flow mechanism is able to synchronize cell and vacuolar surface increase, which is likely the reason why plant cells excel in the speed of cellular expansion.

# Materials and methods

## Plant material and growth conditions

Experiments were carried out in *Arabidopsis thaliana* (Col-0 ecotype). The following plant lines were described previously: *vti11* (zigzag) (**Yano et al., 2003**), *pUBQ10::VAMP711-YFP* (WAVE9Y) (**Geldner et al., 2009**), *pUBQ10::pHGFP-VTI11* (**Takemoto et al., 2018**), *pSYP22::SYP22-GFP* in *syp22* background (**Uemura et al., 2010**), *pBRI1::BRI1-GFP* (**Kleine-Vehn et al., 2008**), *pPGP19::PGP19-GFP* (**Dhonukshe et al., 2008**), *pUBQ10::NPSN12-YFP* (WAVE131Y) (**Geldner et al., 2009**), *CLC-GFP* (**Ito et al., 2011**), *p35S::Lifeact-venus* (**Era et al., 2009**), *pUBQ10::GOT1-YFP* (WAVE18Y) (**Geldner et al., 2009**), *pVHAa1::VHAa1-GFP* (**Dettmer et al., 2006**), *pVHAa3::VHAa3-GFP* (**Dettmer et al., 2006**), *pUBQ10::RABF2a-YFP* (WAVE7Y) (**Geldner et al., 2009**), *p35S::SYP21-GFP* (**Robert et al., 2008**), *amiR vps3* (**Takemoto et al., 2018**), *amiR vps39* (**Takemoto et al., 2018**), *pUBQ10::YFP-VAMP711 amiR vps3* (**Takemoto et al., 2018**). *vti11 pUBQ10::YFP-VAMP711* was obtained by crossing. Seeds were stratified at 4°C for 2 days in the dark and were grown on vertically orientated ½ strength Murashige and Skoog (MS) medium plates containing 1% sucrose under a long-day regime (16 hr light/8 hr dark) at 20–22°C.

## Chemicals

All chemicals were dissolved in dimethyl sulfoxide (DMSO) (Duchefa, NL, cat. no. D1370) and applied in solid or liquid ½ MS medium. FM4-64 (cat. no. T3166) was obtained from Invitrogen/Thermo Fisher Scientific (Waltham, MA), PI (cat. no. 81845) and DEX (cat. no. D1756) from Sigma (St Louis, MO), 2′,7′-bis(2-carboxyethyl)-5(6)-carboxyfluorescein acetoxymethyl ester (BCECF-AM) (cat. no. 51012) from Biotium (Fremont, CA), Wortmannin (WM) (cat. no. HY-10197) from MedChemExpress (Princeton, NJ) and BFA (cat. no. J62340) from Alfa Aesar (Tewksbury, MA). VAC1 (*N′*-[(2,4-dibromo-5-hydroxyphenyl)methylidene]-2-phenylacetohydrazide, ID 5326213), VAC1A1 (2-phenyl-*N'*-(2,4,6-tribromo-3-hydroxybenzylidene)acetohydrazide, ID 5326215), VAC1A2 (*N'*-(2-bromo-5-hydroxybenzylidene)-2-phenylacetohydrazide, ID 5326212), VAC1A3 (*N'*-(2,4-dibromo-5-hydroxybenzylidene)-2-(1-naphthyl)acetohydrazide, ID 5575792), VAC1A4 (*N'*-(2,4-dibromo-5-hydroxybenzylidene)-2-hydroxy-2-phenylacetohydrazide, ID 5568198), and VAC1A5 (*N'*-(3,5-dibromo-4-hydroxybenzylidene)-2-phenylacetohydrazide, ID 5326129) were obtained from ChemBridge/Hit2Lead (San Diego, CA) (https://www.hit2lead.com/search.asp?db=SC).

## VAC1

VAC1 used in this study was either obtained from ChemBridge/Hit2Lead (San Diego, CA) (https://www.hit2lead.com/search.asp?db=SC, ID 5326213) or synthesized in-house as described in VAC1 synthesis (see below). We performed LC-MS analysis to determine the similarity of synthesized and commercially available VAC1. HPLC chromatograms and MS spectra of the synthesized VAC1 were identical to the ones of the reference VAC1. The purity of the synthesized VAC1 (99.2%) was higher than that of the purchased VAC1 (98.2 %). The IUPAC (International Union of Pure and Applied Chemistry) nomenclature was adapted to the VAC1 molecule (*N′*-[(2,4-dibromo-5-hydroxyphenyl)methylidene]-2-phenylacetohydrazide) throughout the manuscript, however an alternative name for VAC1 (*N'*-(2,4-dibromo-5-hydroxybenzylidene)-2-phenylacetohydrazide) is used by the manufacturer Chembridge/Hit2Lead.

## VAC1 synthesis

**Scheme 1.** Preparative route to VAC1. Reaction conditions: (a) EtOH, cat. Fe$_2$(SO$_4$)$_3$/H$_2$SO$_4$, reflux, 3.5 hr, 87%; (b) N$_2$H$_4$·nH$_2$O (50–60% N$_2$H$_4$), MeOH, rt, 24–72 hr, 68%; (c) Br$_2$ (2.12 eq.), DCM, rt, 22 hr, 64%; (d) EtOH, reflux, 3 hr, 76%.

## Experimental

Starting reagents 3-hydroxybenzaldehyde **4** and hydrazine hydrate were purchased from Sigma-Aldrich (cat. no. H19808 and 225819, respectively) and PAA **1** was purchased from Merck (cat. no. P16621). These reagents were used as received without further purification.

Synthesis of **5** was accomplished by following the respective procedure in *Kallman et al., 2014*, and **2** was prepared adapting *Liang et al., 2004*.

TLC was run on Silica 60 glass plates (Merck). Low-resolution mass spectra were obtained on an Agilent 6340 Ion Trap instrument. For m/z values, the most abundant isotope of the isotope distribution is reported.

## Ethyl phenylacetate (2)

A solution of **1** (4.08 g, 30 mmol) in abs. ethanol (150 ml) containing 50 mg of equimolar mixture of iron (III) sulfate and concentrated sulfuric acid was held at reflux for 3.5 hr until no starting material was visible on TLC. Ethanol was distilled off and the residue was dissolved in DCM, washed with saturated aqueous solution of sodium bicarbonate, dried with anh. magnesium sulfate, stripped of solvent and purified by distillation in vacuo (bp 105–108°C at 13 Torr) to afford **2** as colorless liquid. Yield: 4.3 g (87%).

## PAA hydrazide (3)

Hydrazine hydrate (2.81 ml, 45–54 mmol) was added dropwise to a solution of **2** (4.0 g, 24 mmol) in methanol (28 ml) and the mixture was agitated at rt for 24–72 hr until complete consumption of the ester (TLC). Excess solvent was removed by distillation and, upon cooling, the residue crystallized as long thin needles. The mass was crushed, washed on a glass filter with cold water, and recrystallized from 90% aq. ethanol. Yield: 3.0 g (82%) before and 2.5 g (68%) after recrystallization.

## 2,4-Dibromo-5-hydroxybenzaldehyde (5)

To a solution of **4** (1.0 g, 8.2 mmol) in DCM (10 ml) was slowly added elemental bromine (2.78 g 17.4 mmol) and the reaction mixture was stirred at rt for 22 hr. The reaction progress was monitored by HPLC. Upon completion, the reaction was quenched by dropwise addition of 15% aq. sodium thiosulfate (4.8 ml), stirred at rt for 1 hr and filtered. The filter cake was washed with water (2 × 5 ml) and dissolved in acetic acid (6.4 ml) at 95°C. After cooling to 50°C, water (3.4 ml) was added dropwise to a stirred solution, resulting in crystallization of the product. The crystals were stirred for 4 hr at 15°C, filtered, washed with water (2 × 5 ml) and dried. Yield: 1.68 g (73%). The product (pink crystals) was

additionally purified by vacuum sublimation, affording **5** as white crystalline powder (1.48 g, 64%). m/z (ESI⁻): 278.8 [M-H]⁻.

## N'-[(2,4-dibromo-5-hydroxyphenyl)methylidene]-2-phenylacetohydrazide (VAC1)

A mixture of **3** (0.53 g, 3.53 mmol) and 1.02 eq. of **5** in 9.6 ml of ethanol was held at reflux for 3 hr until complete consumption of the starting material (TLC). Upon cooling, a copious white precipitate of the title compound was formed. It was filtered, washed with petroleum ether, and dried. The final product was additionally purified by recrystallization from ethanol. Yield 1.1 g (76%). m/z (ESI⁻): 410.9 [M-H]⁻.

## UHPLC-UV-(SIR)MS

To determine the conversion of VAC1 into free PAA, 6-day-old seedlings were treated with DMSO or 10 µM VAC1 for 2.5 hr in liquid medium and then flash-frozen in liquid nitrogen. The extraction and quantification methods for both analytes were optimized based on the previously described method (*Pařízková et al., 2021*). Briefly, the plant samples (~30 mg FW) were extracted into 100% methanol and purified using liquid-liquid extraction method into 900 µl of extraction solution (methanol:$H_2O$:hexane – 1:1:1), evaporated and dissolved in 50 µl of 100% methanol. Two µl of each sample were injected onto the reversed-phase column (Kinetex C18 100A, length 50 mm, diameter 2.1 mm, particle size 1.7 µm; Phenomenex, Torrance, CA) and analyzed by an Acquity UPLC H-Class System (Waters, Milford, MA) coupled with an Acquity PDA detector (scanning range 190–400 nm with 1.2 nm resolution) and a single quadrupole mass spectrometer QDa MS (Waters MS Technologies, Manchester, UK) equipped with an electrospray interface (ESI). The analytes were eluted from the column within 9-min-linear gradient of 10:90 to 95:5 A:B using 0.1% acetic acid in methanol (A) and 0.1% acetic acid in water (B) as mobile phases at a flow rate of 0.5 ml·min⁻¹ and column temperature of 40°C. After every analysis, the column was washed with 95% methanol and then equilibrated to initial conditions (1.0 min). Both analytes were detected by selected ion recording (SIR) using the positive and negative electrospray modes (ESI⁺ and ESI⁻) as follows: VAC1 detected as [M+H]⁺, m/z 411.0, and PAA as [M-H]⁻, m/z 135.0. The MS settings were optimized as follows: source temperature, 120°C; desolvation temperature, 600°C; capillary voltage, 0.8 kV. Chromatograms were processed by MassLynx V4.2 software (Waters) and quantification was performed from external calibration using a recovery factor. Additionally, M1-M3 metabolites of VAC1 were detected using a full-scan mode (m/z 50–1000) operated in ESI⁻ with post-data acquisition extraction of ion chromatograms for m/z 425.0, 571.0, and 613.0 for M1, M2, and M3, respectively. To determine the distribution of VAC1 metabolites, the UV chromatograms were extracted for $\lambda_{max}$ (291 nm) and the percentage representation of metabolites in plant extracts was determined by the integration of peak areas in respective retention times (M1, 4.37 min; M2, 5.26 min; M3, 5.39 min; VAC1, 6.65 min).

## UHPLC-HRMS

Samples of DMSO- and VAC1-treated plants were prepared as described above (UHPLC-SIR-MS). Plant extracts (2 µl) were injected onto a Kinetex C18 column and separated using chromatographic conditions as described above. The HRMS analysis was achieved by a hybrid Q-TOF tandem mass spectrometer Synapt G2-Si (Waters MS Technologies) as described previously (*Buček et al., 2018*). Briefly, the effluents were introduced into the HRMS instrument (ESI⁻; capillary voltage, 0.75 kV; source offset, 30 V; desolvation/source temperature, 550/120°C; desolvation/cone gas flow, 1000/50 l·hr⁻¹; LM/HM resolution, 2.8/14.75; ion energy 1/2, 0.5/1.0 V; entrance/exit voltages, 0.5 V; collision energy, 6 eV). The determination of exact mass was performed by the external calibration using lock spray technology and a mixture of leucine/encephalin (1 ng·µl⁻¹) in an acetonitrile and water (1:1) solution with 0.1% formic acid as a reference. Data acquisition was performed in full-scan mode (50–1000 Da) with a scan time of 0.5 s and all analytes were detected as [M-H]⁻ in the MS spectrum. All data were processed using MassLynx 4.1 software (Waters). The accurate masses of VAC1 metabolites were calculated and then used to determine the elemental composition of the analytes with a fidelity ranging from 1.6 to 2.6 ppm. The VAC1 metabolites were identified based on correlation of the theoretical monoisotopic weights of deprotonated forms [M-H]⁻ and detected accurate masses of each precursor of M1-M3 as well as based on the presence of two bromine stable isotopes (79Br and 81Br) in HRMS spectrum resulting in the characteristic isotope pattern of dibromo derivatives.

## Phenotype analysis

Vacuolar occupancy and vacuole surface area as well as cell surface area and BFA body volume were quantified in 6-day-old seedlings. For 3D reconstructions of cells, confocal images were processed using Imaris (vacuolar occupancy of cells, BFA bodies, vacuole, and cell surface) as described previously (*Dünser et al., 2019*). BCECF staining was performed as described previously (*Scheuring et al., 2015*). For the analysis of main root growth, 6-day-old seedlings were used, unless indicated otherwise. For the analysis of main root growth of amiR *vps3* and amiR *vps39* seedlings were grown for 4 days before transfer to DMSO, DEX, VAC1, or DEX + VAC1 and then grown for additional 3 days. Plates were scanned and root length assessed using ImageJ. Selection of pUBQ10::YFP-VAMP711 amiR *vps3* seedlings for confocal microscopy analysis was done as described previously (*Takemoto et al., 2018*).

## Confocal microscopy

For image acquisition, a Leica TCS SP5 (DM6000 CS) or a Leica SP8 (DMi8) confocal laser scanning microscope, equipped with a Leica HCX PL APO CS 63×1.20 water-immersion objective, was used. GFP and BCECF were excited at 488 nm (fluorescence emission: 500–550 nm), YFP and FM4-64 at 514 nm (fluorescence emission YFP: 525–578 nm; fluorescence emission FM4-64: 670–790 nm), and PI at 561 nm (fluorescence emission: 644–753 nm). Roots were mounted in PI solution (0.02 mg/ml) for the counterstaining of cell walls. Z-stacks were recorded with a step size of 420 nm (for 3D reconstructions) or 1.5 μm (for maximum projections of VAC1 bodies).

## 3D reconstruction of cells

Imaris 8.4.0 and 9.0 were used for the reconstruction of cell, vacuole, and BFA body volumes. Based on the PI channel, every third slice of the z-stack was utilized to define the cell borders using the isoline, magic wand, or manual (distance) drawing functions in the manual surface creation tool. After creating the surface corresponding to the entire cell, a masked channel (based on BCECF or FM4-64) was generated by setting the voxels outside the surface to 0. Subsequently, a second surface (based on the masked BCECF or FM4-64 channel) was generated automatically with the smooth option checked. Surface detail was set to identical values within each region (early and late meristem and early and late elongation zone) to ensure comparability of obtained surface area values. The obtained surface was visually compared to the underlying BCECF or FM4-64 channel and, if necessary, the surface was fitted to the underlying signal by adjusting the absolute intensity threshold slider. Finally, volumes and surface areas of both surfaces were extracted from the statistics window.

## Sample preparation for electron microscopy

Around 3 mm of the *Arabidopsis* root tips were sectioned and immediately submerged in fixative solution containing 2.5% glutaraldehyde and 4% p-formaldehyde in MTSB buffer. Samples were vacuum infiltrated for 15 min at room temperature, fixed for 4 hr at room temperature, and kept in fixative overnight at 4°C. After five times washing (10 min each) in MTSB buffer, samples were post-fixed in 1% osmium tetroxide in $H_2O$ on ice for 4 hr with shaking. Samples were washed with water four times (5 min each), and in bloc stained for 2 hr with 2% uranyl acetate in water. After washing three times with water (5 min each), samples were dehydrated in EtOH graded series (30%–50%–70%–80%–90%–95%, 15 min each). Fully dehydration was achieved by incubating the samples twice in 100% EtOH for 30 min followed by incubation in acetone twice (30 min each). Samples were gradually embedded in resin by incubating them in acetone: Agar 100 mixtures (3:1, 1:1, 1:3–8 hr each) and finally in pure Agar 100 resin three times (8 hr each). The root tips were flat embedded according to *Kolotuev, 2014*, and the resin was polymerized overnight at 60°C. The flat embedded samples were mounted on resin blocks and 70 nm longitudinal sections of root-tips were obtained with a Reichert-Jung Ultracut-E ultramicrotome and collected on slot grids. Sections were contrasted with 2% uranyl acetate and lead citrate according to *Reynolds, 1963*, and examined in a Hitachi 7800 TEM (100 kV) equipped with an EMSIS Xarosa camera.

## Small molecule screen

Roughly 20 pUBQ10::YFP-VAMP711 seeds were germinated in 12-well plates with each well containing 1.5 ml liquid ½ MS+ supplemented with solvent control (DMSO) or 50 μM of small molecules.

Four-day-old seedlings were screened for enhanced fluorescence signal using a fluorescence binocular. Twelve hit compounds were further examined using confocal microscopy (see *Figure 2—figure supplement 1* and *Figure 2—figure supplement 2*). The small molecules used for the primary screen were identified as inhibitors of tobacco pollen germination (*Drakakaki et al., 2011*).

### Quantification of VAC1 and BFA bodies

Maximum z-projections of three to eight slices (step size 1.5 μm) and an ROI of 5 μm × 5 μm were used for quantification of the FM4-64 fluorescence signal. Four to ten roots were quantified with three bodies per root.

### Quantification of intracellular BRI1-GFP fluorescence signal

Maximum z-projections of three to six slices (step size 1.5 μm) and an ROI of 10 μm × 5 μm were used for quantification of the intracellular BRI1-GFP fluorescence signal. Eight roots were quantified, using two early meristematic and two late meristematic cells per root.

### Co-localization analysis

Quantification of co-localization was done using ImageJ. Two to three atrichoblast cells in the specified regions were used per root. The area was marked and cropped (Image > Crop), subsequently the surrounding of the cell was cleared (Edit > Clear outside) and the channels were split (Image > Color > Split channels). Co-localization (Analyze > Co-localization > Co-localization threshold) was analyzed and the values for Rtotal as well as the scatter plots were extracted.

### Time course experiments

Seedlings were mounted on agar blocks containing solvent control DMSO or VAC1. The agar blocks were placed in chamber slides and subsequently installed on an inverted confocal microscope (Leica SP8). Z-stacks (three to five slices, step size: 1.5 μm) were acquired every 10 min and maximum z-projections used for quantification of root tip displacement and cell length.

### Cell length analysis

Six-day-old seedlings were used for cell length measurements of differentiated atrichoblasts (*Figure 1C and D*). Fully differentiated cells were identified as described previously (*Löfke et al., 2015*).

### Statistical analysis and reproducibility

All graphs were generated using GraphPad Prism 9. Each experiment was repeated at least twice independently with similar results. The depicted data are the results from one representative experiment.

## Acknowledgements

We are grateful to N Geldner, S Robert, K Schumacher, and T Ueda for sharing published material. The library of 360 bioactive small molecules was kindly provided by N Raikhel. We thank Maximilian Kaiser for helpful discussions about VAC1 and the BOKU-VIBT Imaging Center as well as the ZBSA-Life Imaging Center for access. We would like to thank Rosula Hinnenberg of the EM facility at the Faculty of Biology, University of Freiburg, for her assistance during the generation of EM data. The TEM (Hitachi HT7800) was funded by the DFG grant (project number 426849454) and is operated by the University of Freiburg, Faculty of Biology, as a partner unit within the Microscopy and Image Analysis Platform (MIAP) and the Life Imaging Center (LIC), Freiburg.

This work was supported by the Ministry of Education, Youth and Sports of the Czech Republic (European Regional Development Fund-Project 'Plants as a tool for sustainable global development' No. CZ.02.1.01/0.0/0.0/16_019/0000827 to ON), The Austrian Academy of Sciences (ÖAW) (DOC fellowship to KD), Austrian science fund (FWF; P 33044 to JK-V), German Science fund (DFG; 470007283 and CIBSS – EXC-2189 to JK-V), and the European Research Council (ERC; 639478-AuxinER to JK-V).

# Additional information

## Competing interests

Jürgen Kleine-Vehn: Senior editor, eLife. The other authors declare that no competing interests exist.

## Funding

| Funder | Grant reference number | Author |
| --- | --- | --- |
| Austrian Science Fund | P 33044 | Jürgen Kleine-Vehn |
| Deutsche Forschungsgemeinschaft | 470007283 | Jürgen Kleine-Vehn |
| CIBSS | EXC-2189 | Jürgen Kleine-Vehn |
| European Research Council | 639478-AuxinER | Jürgen Kleine-Vehn |
| Ministry of Education, Youth and Sports of the Czech Republic (European Regional Development Fund-Project "Plants as a tool for sustainable global development" | CZ.02.1.01/0.0/0.0/16_019 /0000827 | Ondřej Novák |
| Deutsche Forschungsgemeinschaft | 426849454 | Marta Rodriguez-Franco |
| Austrian Academy of Sciences | DOC fellowship | Kai Dünser |

The funders had no role in study design, data collection and interpretation, or the decision to submit the work for publication.

## Author contributions

Kai Dünser, Conceptualization, Data curation, Funding acquisition, Investigation, Supervision, Validation, Visualization, Writing – original draft, Writing – review and editing; Maria Schöller, Data curation, Formal analysis, Investigation, Validation, Visualization; Ann-Kathrin Rößling, Data curation, Investigation, Validation, Visualization, Writing – review and editing; Christian Löfke, Data curation, Formal analysis, Investigation, Supervision, Validation, Visualization; Nannan Xiao, Investigation, Validation; Barbora Pařízková, BP conducted UHPLC-SIR-MS and UHPLC-HRMS., Data curation, Formal analysis, Investigation, Validation, Visualization, Writing – review and editing; Stanislav Melnik, Investigation, SM synthesised VAC1., Validation, Visualization, Writing – review and editing; Marta Rodriguez-Franco, Investigation, MR-F conducted transmission electron microscopy., Validation, Visualization, Writing – review and editing; Eva Stöger, Resources, Writing – review and editing; Ondřej Novák, Data curation, Formal analysis, Funding acquisition, Investigation, ON conducted UHPLC-SIR-MS and UHPLC-HRMS., Resources, Supervision, Validation, Visualization, Writing – review and editing; Jürgen Kleine-Vehn, Conceptualization, Formal analysis, Funding acquisition, Methodology, Resources, Supervision, Validation, Writing – original draft, Writing – review and editing

## Author ORCIDs

Kai Dünser ⓘ http://orcid.org/0000-0002-8974-2072
Jürgen Kleine-Vehn ⓘ http://orcid.org/0000-0002-4354-3756

## Decision letter and Author response

Decision letter https://doi.org/10.7554/eLife.75945.sa1
Author response https://doi.org/10.7554/eLife.75945.sa2

# Additional files

## Supplementary files

• Transparent reporting form

## Data availability
Source Data file contains the numerical data used to generate the figures.

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
