## [Editor Report]

Plant cells can grow to extraordinarily large volumes; Arabidopsis root cells, for example, can expand beyond 50um long. Vacuole expansion is correlated with cell elongation, presumably to "fill up" the volume of the cell without requiring a tremendous volume of cytoplasm. Here, the authors carefully characterize an exciting new small molecule inhibitor of endocytic trafficking to the vacuole. This new tool will be valuable to researchers studying endocytic trafficking and vacuole biogenesis in plants.

---

## [Decision Letter]

**Decision letter after peer review:**

Thank you for submitting your article "Endocytic Trafficking Promotes Vacuolar Enlargements for Fast Cell Expansion Rates in Plants" for consideration by *eLife*. Your article has been reviewed by 3 peer reviewers, one of whom is a member of our Board of Reviewing Editors, and the evaluation has been overseen by Detlef Weigel as the Senior Editor. The following individual involved in review of your submission has agreed to reveal their identity: Takashi Ueda (Reviewer #3).

Essential revisions:

The reviewers agree that the paper will be interesting to a wide audience, but have highlighted three key points that are essential to address with revisions:

1. Conduct live cell imaging of endocytic markers (e.g. CME components) and/or an additional cargo to support the claim that endocytic rate is different between meristematic and elongating cells, according to suggestions made by Reviewers 1 and 2.

2. Revise the discussion around Figures 5 and 6 to clarify why cell surface area is being compared to BFA body volume (as Reviewer 2 suggests) and/or provide a more rigorous analysis (as Reviewers 1 and 3 suggest).

3. Revise the results and discussion surrounding the WM experiment as indicated by Reviewers 2 and 3. An additional PI3K inhibitor (e.g. LY294002) and/or treatment with a PI4K inhibitor (e.g. PAO) as a negative control would further support these claims.

The reviewers have also raised a number of additional points that are worth consideration, and their reviews are included below.

*Reviewer #1 (Recommendations for the authors):*

1. From the perspective of cell wall synthesis, it's unsurprising that increased endocytic trafficking is correlated with cell expansion since the volume of the cell wall must increase much greater than the surface area of the plasma membrane during elongative growth. I think the discussion could be improved by incorporating this idea, e.g. the work of Anne-Mie Emons (Ketelaar et al. 2008).

2. FM4-64 quantification is difficult because it is challenging to ensure that uptake is equal across cells. For example, I am surprised that so many cells in the meristematic zone showed R = 0 (data points at the bottom of the boxes in Figure 5I), indicating that even after 2 hours, no FM4-64 has reached the tonoplast. This is especially puzzling because the authors have presented images of meristematic cells with clear FM4-64 labelling of the tonoplast in other figures (e.g. Figure 3A, Figure 5G). What do the authors speculate is happening in these cases?

3. Does the difference in "endocytic rate" implied by more intense FM4-64 staining and larger BFA bodies in the elongation zone, compared to the meristematic zone really imply more endocytosis? Isn't there also more FM4-64 available since there is more surface area to the cell? Then the BFA bodies might appear brighter or larger simply because there is more dye in them? This claim probably requires secondary evidence, such as quantification of density and dynamics of a clathrin marker in these different cell types, or pulse-chase of a marker (e.g. photoactivatable or photoconvertible). In particular, the authors' hypothesis makes an easily-testable prediction: that endocytic events should be more frequent at the PM in elongation-zone cells, compared to the meristematic zone, even after accounting for the change in surface area. The authors already have a CME marker, so this is a straight-forward experiment for them to conduct.

4. It is confusing to compare surface area of a large object (the cell) to volume of a small object (the BFA body) as in Figure 5F. There is a crossover point in surface area-to-volume ratios at which the ratio is >1 for a small object and <1 for a large object and I think this graph oversimplifies this important point. The volume of the BFA body is small relative to the surface area of the cell, so the FM4-64 signal in the BFA body is concentrated, compared to the signal at the cell surface. I suggest the authors present these data some other way (e.g. plot surface area vs volume for BFA bodies and cells at different stages) to better illustrate this. I also think that the results and discussion surrounding this figure should be revised to clearly articulate this issue.

5. For the wide readership of *eLife*, it would be great to see whether VAC1 has any activity on vacuolar shape in other plants, or whether it has any effects on the endomembrane system in mammalian or fungal cells.

6. After introducing the vti11 phenotype, the authors conclude that "compensatory mechanisms" ensure vacuolar occupancy of the cell and "safeguard cell expansion" however vti11 vti12 double mutants are lethal, indicating that genetic redundancy, not compensatory mechanisms, are at work. Please correct.

*Reviewer #2 (Recommendations for the authors):*

– A marker-assisted comparison between VAC1 and Concanamycin A might be useful here as the latter compound also interferes with vacuolar membrane fusion, albeit likely less specific.

– For some of the cell types analyzed in Figure 1F, the numbers are a bit low. Some additional quantifications could be performed here to strengthen the data.

– On line 89, I would rephrase the striking effect of VAC1 on vacuolar morphology as several other VACs have different, yet equally striking effects.

– The viability of the cells upon treatment with VAC1A4 would be better addressed by FDA rather than with PI.

– On line 129, as it was not tested, the authors could include the possibility that the alternative compounds do not have a similar effect because they might be less stable that the original VAC1.

– On line 134, the statement that VAC1 interferes with the localization of the SNARE complex is odd. It affects vacuolar morphology, hence obviously it affects vacuolar SNARE localization. The authors might want to rephrase this by saying that VAC1 does not specifically affect one of the components of the SNARE complex as they observe similar localizations for various subunits of this complex.

– On line 136, it is strange to use Scheuring et al., 2015 as initial reference for the styryl dye FM4-64.

– On line 144, the assessment of the trafficking of PM resident proteins to the vacuole might be better visualized using dark-treatment to visualize differential vacuolar GFP accumulation.

– On line 150, the conclusion that VAC1 does not affect endocytic trafficking or actin dynamics could be better addressed using dynamic imaging e.g. using TIRF or via rainbow-color overlays of different timepoints of confocal slices for example.

– In Figure 3F, optimally, the quantifications could be normalized instead of measured by mean grey values. The PM signal appears very different between VAC1 and VAC1 + BFA. Also, the results of VAC1 + Wm are puzzling, the PM staining of FM in panel G is very different from the one in panel E and the disappearance of BRI1 from the TGN upon the combined treatment of VAC1 and Wm is not addressed. For clarity, I would remove the Wm data as the observations cannot be accurately explained for the moment and no clear conclusion can be drawn from this.

– In supplemental Figure 5, it is unclear to me why the emphasis lies on transversal PM.

– On line 217 and following, the authors propose enhanced endocytic trafficking correlating with elongation based on the size of VAC1 compartments and BFA bodies, which are larger in elongating cells versus meristematic cells. This conclusion assumes that there are no differences between these cells in terms of compound penetrance or the differential activity of trafficking pathways other than endocytic that could affect the size of the observed bodies. This assumption could be mentioned/discussed. In order to provide stronger support for their claim that endocytic trafficking is enhanced in elongating cells versus meristematic cells, time lapse imaging of roots using PM resident markers transiently expressed via a heat shock promotor or constitutively expressed following CHX could be considered.

– On line 229-230, the authors might want to explain better their choice to measure area instead of volume for the non-expert reader.

*Reviewer #3 (Recommendations for the authors):*

Figure 3G. Wm also inhibits an early event of endocytosis at the PM, probably through inhibition of production of PI4P and/or PI45P2 in the plasma membrane. The result presented in Figure 3G and H should be interpreted taking this effect of WM into authors' account.

Figure 4C, G, and S4. The definition of "relative growth" is unclear. Is this a relative value of root lengths to the mean value of root length of control plants? If this is the case, variation in the control plants is not correctly reflected in the data. Authors should interpret the results based on Figure 4B and Figure 4G?

Figure 5A-D. It would be not appropriate to compare just the mean grey value to see an endocytic activity. The mean grey value could be higher in larger cells with a larger area of the PM even if the endocytic rate is constant between these cell types.

---

## [Author Response]

Reviewer #1 (Recommendations for the authors):1. From the perspective of cell wall synthesis, it's unsurprising that increased endocytic trafficking is correlated with cell expansion since the volume of the cell wall must increase much greater than the surface area of the plasma membrane during elongative growth. I think the discussion could be improved by incorporating this idea, e.g. the work of Anne-Mie Emons (Ketelaar et al. 2008).

We thank you for this comment. We personally reached out to the first author of the mentioned publication and discussed this aspect. Based on the published assumption, plants have a substantial need for membrane trafficking at the plasma membrane, which is likely maintained by membrane recycling between the plasma membrane and the trans golgi network/early endosome (facilitated by constitutive endocytosis and exocytosis). Accordingly, vesicle sorting to the vacuole drains membranes away from the recycling pathway, possibly affecting not only surface enlargements, but also cell wall synthesis during cellular elongation. We thank you for raising this aspect and we have modified the discussion along these lines.

2. FM4-64 quantification is difficult because it is challenging to ensure that uptake is equal across cells. For example, I am surprised that so many cells in the meristematic zone showed R = 0 (data points at the bottom of the boxes in Figure 5I), indicating that even after 2 hours, no FM4-64 has reached the tonoplast. This is especially puzzling because the authors have presented images of meristematic cells with clear FM4-64 labelling of the tonoplast in other figures (e.g. Figure 3A, Figure 5G). What do the authors speculate is happening in these cases?

We indeed propose that FM4-64 uptake is not equal along the roots, showing enhancement at the onset of elongation. The initial FM4-64 labelling of the plasma membrane was relatively constant along the root zonation (see Figure 5—figure supplement 1 A-B).

The degree of FM4-64 labeling may slightly fluctuate in individual cells (at/before onset of elongation) and experimental conditions, but was overall consistent in all our experiments. We reproducibly observe enhanced FM4-64 accumulation in BFA and VAC1 bodies in elongating cells. This effect could relate to regulations of endocytosis at the plasma membrane and/or altered membrane sorting at the trans golgi network. Both endocytosis and endosomal sorting independently contribute to the rate of endocytic membrane trafficking to the vacuole. We feel this aspect was unclear and we improved the discussion on this matter in the revised version of our manuscript.

Figure 3A display epidermal cells at the onset of elongation (note that cells are slightly longer than wide) in the transition zone of wild type seedlings, displaying FM4-64 labeling at the tonoplast. Figure 5G shows FM4-64 labeling in the transgenic pUBQ10:YFP-VAMP711 line, which may differ in some aspects compared to wild type, but does allow the assessment of colocalization. While most cells show some degree of FM4-64 at the tonoplast in late meristematic cells, we indeed captured early meristematic cells with less FM4-64 tonoplast labeling (Figure 5G). In contrast, elongating cells always showed a high degree of colocalization.

3. Does the difference in "endocytic rate" implied by more intense FM4-64 staining and larger BFA bodies in the elongation zone, compared to the meristematic zone really imply more endocytosis? Isn't there also more FM4-64 available since there is more surface area to the cell? Then the BFA bodies might appear brighter or larger simply because there is more dye in them? This claim probably requires secondary evidence, such as quantification of density and dynamics of a clathrin marker in these different cell types, or pulse-chase of a marker (e.g. photoactivatable or photoconvertible). In particular, the authors' hypothesis makes an easily-testable prediction: that endocytic events should be more frequent at the PM in elongation-zone cells, compared to the meristematic zone, even after accounting for the change in surface area. The authors already have a CME marker, so this is a straight-forward experiment for them to conduct.

We thank you for raising this point, which certainly helped us to improve the description and discussion of our work. We realized that the term endocytic trafficking towards the vacuole can be misleading. The degree of membrane flow from the plasma membrane to the vacuole indeed depends on endocytosis, but is also defined by sorting endosomes, which gate recycling of membranes back to the plasma membrane and further endocytic membrane trafficking to the vacuole. Accordingly, enhanced endocytic trafficking to the vacuole could be achieved independent of the endocytosis rates. We have improved the discussion on this matter and hope to have thereby clarified this aspect. While our work suggests a reprogramming of vesicle trafficking events at the onset of elongation, upcoming work should in depth address the underlying molecular mechanism. Considering its potential complexity, this aspect is however beyond the scope of this manuscript.

If the endocytosis rate remains constant, a cellular surface increase could indeed lead to higher uptake of endocytic dyes. The elucidation of the cellular surface is a read-out for the amount of plasma membranes. In contrast, a BFA compartment is filled with secretory vesicles (with a diameter of below 0,1 µm; see Geldner et al., 2001) and hence the BFA body surface insufficiently quantifies these membranes. We hence addressed both the relative increase in cellular surface and volume between early and late meristematic cells. While these cells roughly doubled in size, particularly the BFA volume showed a much stronger relative increase. Accordingly, we assume that the increase in cell surface does not recapitulate the increases in vacuolar membrane trafficking, suggesting that vesicle trafficking undergoes reprogramming during cellular elongation. We however certainly do not rule out that the surface increase contributes to this mechanism. We improved the depiction and discussion of this set of data.

4. It is confusing to compare surface area of a large object (the cell) to volume of a small object (the BFA body) as in Figure 5F. There is a crossover point in surface area-to-volume ratios at which the ratio is >1 for a small object and <1 for a large object and I think this graph oversimplifies this important point. The volume of the BFA body is small relative to the surface area of the cell, so the FM4-64 signal in the BFA body is concentrated, compared to the signal at the cell surface. I suggest the authors present these data some other way (e.g. plot surface area vs volume for BFA bodies and cells at different stages) to better illustrate this. I also think that the results and discussion surrounding this figure should be revised to clearly articulate this issue.

In the revised version, we show now the relative surface and volume increase for the cell and for the BFA compartment. In both cases, the relative increase of the BFA compartment (surface and volume) exceeds the relative size increase of the cell. Considering the accumulation of the secretory vesicles in the BFA compartment, we propose that the relative increase in BFA compartment size is not a mere consequence of cell size increase. We improved the discussion surrounding this finding in the revised version of the manuscript.

5. For the wide readership of eLife, it would be great to see whether VAC1 has any activity on vacuolar shape in other plants, or whether it has any effects on the endomembrane system in mammalian or fungal cells.

We are indeed interested in testing VAC1 in other plants, mammalian and fungal cells and also have ongoing collaborations with other labs on this topic. However, this aspect needs to be thoroughly addressed and is beyond the scope of this manuscript.

6. After introducing the vti11 phenotype, the authors conclude that "compensatory mechanisms" ensure vacuolar occupancy of the cell and "safeguard cell expansion" however vti11 vti12 double mutants are lethal, indicating that genetic redundancy, not compensatory mechanisms, are at work. Please correct.

We agree that VAC1 likely overcomes genetic redundancy. On the other hand, we do not know whether the roundish vacuoles in *vti11* mutants are due to redundant *VTI12* function or whether this relates to other compensatory mechanisms. We, however, followed your suggestion and simplified our statement.

Reviewer #2 (Recommendations for the authors):– A marker-assisted comparison between VAC1 and Concanamycin A might be useful here as the latter compound also interferes with vacuolar membrane fusion, albeit likely less specific.

A similar analysis has been performed (see Dettmer et al., 2006) and FM4-64 + ConcA looks quite different from FM4-64 + VAC1. In contrast to VAC1, ConcA also impacts on the integrity of the trans golgi network.

– For some of the cell types analyzed in Figure 1F, the numbers are a bit low. Some additional quantifications could be performed here to strengthen the data.

Detailed z-stack imaging and 3D reconstructions of different cell types are unfortunately very elaborate and time consuming. It is therefore hardly feasible to reach a very high sample size during a single imaging session. The data was highly reproducible in our independent experiments. Instead of pooling distinct biological replicates, we preferred to show a representative experiment with a sample size between 7 and 16 reconstructed cells for each category.

– On line 89, I would rephrase the striking effect of VAC1 on vacuolar morphology as several other VACs have different, yet equally striking effects.

We agree with your suggestion and rephrased our wording.

– The viability of the cells upon treatment with VAC1A4 would be better addressed by FDA rather than with PI.

Thank you for the suggestion. When characterizing VAC1A4 in more detail, we will certainly consider using fluorescein diacetate as a viability staining in our future experimental set up.

– On line 129, as it was not tested, the authors could include the possibility that the alternative compounds do not have a similar effect because they might be less stable that the original VAC1.

We fully agree. In the revised version of this manuscript, we noted this possibility.

– On line 134, the statement that VAC1 interferes with the localization of the SNARE complex is odd. It affects vacuolar morphology, hence obviously it affects vacuolar SNARE localization. The authors might want to rephrase this by saying that VAC1 does not specifically affect one of the components of the SNARE complex as they observe similar localizations for various subunits of this complex.

VAC1 treatment indeed affects the morphology of the vacuole, but also induces a preferential accumulation of vacuolar SNARE proteins in aster like aggregates. We rephrased and improved the description of this aspect in the revised version of this manuscript.

– On line 136, it is strange to use Scheuring et al., 2015 as initial reference for the styryl dye FM4-64.

We included additional references in the revised version, such as:

Vida and Emr (1995): A new vital stain for visualizing vacuolar membrane dynamics and endocytosis in yeast. *The Journal of Cell Biology*, *128*(5), 779-792.

Ueda, T., Yamaguchi, M., Uchimiya, H., and Nakano, A. (2001). Ara6, a plant-unique novel type Rab GTPase, functions in the endocytic pathway of *Arabidopsis thaliana*. *The EMBO Journal*, *20*(17), 4730-4741.

– On line 144, the assessment of the trafficking of PM resident proteins to the vacuole might be better visualized using dark-treatment to visualize differential vacuolar GFP accumulation.

We indeed have previously established the dark assay to address the turnover of plasma membrane proteins (Kleine-Vehn et al., 2008). However, dark conditions also affect growth responses, which is hence not ideal for research on cell expansion. VAC1 treatments could become hence a useful tool to address the rate of protein trafficking towards the vacuole, which may complement dark treatments.

– On line 150, the conclusion that VAC1 does not affect endocytic trafficking or actin dynamics could be better addressed using dynamic imaging e.g. using TIRF or via rainbow-color overlays of different timepoints of confocal slices for example.

We fully agree that additional work would be needed to address the functionality of endocytosis and actin cytoskeleton in full. We improved the description of this aspect in our revised version.

– In Figure 3F, optimally, the quantifications could be normalized instead of measured by mean grey values. The PM signal appears very different between VAC1 and VAC1 + BFA. Also, the results of VAC1 + Wm are puzzling, the PM staining of FM in panel G is very different from the one in panel E and the disappearance of BRI1 from the TGN upon the combined treatment of VAC1 and Wm is not addressed. For clarity, I would remove the Wm data as the observations cannot be accurately explained for the moment and no clear conclusion can be drawn from this.

Normalization of BFA body signal to PM signal for each experiment could be troubling, because BFA may also affect the signal at the PM as it reduces recycling events. However, we assume that the appearance of discriminable BFA compartments (labelled by FM4-64 only) and the reduced signal intensity of FM4-64 in VAC1 compartments (labelled by FM4-64 and SYP21), is very suggestive that VAC1 functions downstream of the TGN. We agree that the data on Wm is not fully conclusive. We followed your suggestion and removed our Wm related data from the revised version of our manuscript.

– In supplemental Figure 5, it is unclear to me why the emphasis lies on transversal PM.

The quantification of signal intensities of transversal plasma membranes of z-stack maximum projections allows, to our opinion, more accurate/precise measurements of the respective cell type. The signal from longitudinal plasma membranes may contain information from subjacent (cortex) cells, which we hereby exclude.

– On line 217 and following, the authors propose enhanced endocytic trafficking correlating with elongation based on the size of VAC1 compartments and BFA bodies, which are larger in elongating cells versus meristematic cells. This conclusion assumes that there are no differences between these cells in terms of compound penetrance or the differential activity of trafficking pathways other than endocytic that could affect the size of the observed bodies. This assumption could be mentioned/discussed. In order to provide stronger support for their claim that endocytic trafficking is enhanced in elongating cells versus meristematic cells, time lapse imaging of roots using PM resident markers transiently expressed via a heat shock promotor or constitutively expressed following CHX could be considered.

When inspecting meristematic and elongating epidermal cells, VAC1 fully blocked in all inspected cells the FM4-64 uptake into the vacuole. Moreover, we did not detect obvious differences in VAC1 induced accumulation of vacuolar SNAREs. Accordingly, we assume that VAC1 shows similar activity in meristematic and elongating cells. Considering that BFA treatments (Figure 5C-F) as well as drug free assays (Figure 5G-I) had a similar outcome, we assume that drug penetrance is likely of lower importance.

We also followed your suggestion and addressed also the VAC1 sensitivity of plasma membrane marker pBRI1:BRI1-GFP, which also showed more intense BRI1 accumulation in elongating epidermal cells. We included this set of data and improved as suggested the overall discussion of this matter in the revised version of this manuscript.

– On line 229-230, the authors might want to explain better their choice to measure area instead of volume for the non-expert reader.

BFA compartments are comprised of accumulating vesicles and hence the surface of the compartment may not fully capture the amounts of membranes. In the revised version of this manuscript, we now provide the relative increase of the surface as well as the volume for both the cell and BFA compartment. We improved the data depiction and overall discussion on this matter. For further details, please, also see response to Reviewer 1.

Reviewer #3 (Recommendations for the authors):Figure 3G. Wm also inhibits an early event of endocytosis at the PM, probably through inhibition of production of PI4P and/or PI45P2 in the plasma membrane. The result presented in Figure 3G and H should be interpreted taking this effect of WM into authors' account.

We agree with your concerns, which were also raised by reviewer 2. Considering that the interpretation of the data is troubling, we followed the suggestion of Reviewer 2 and removed the data in the revised version of this manuscript.

Figure 4C, G, and S4. The definition of "relative growth" is unclear. Is this a relative value of root lengths to the mean value of root length of control plants? If this is the case, variation in the control plants is not correctly reflected in the data. Authors should interpret the results based on Figure 4B and Figure 4G?

Thank you for the comment. We adapted this analysis and displayed the variation of control plants in the revised version.

Figure 5A-D. It would be not appropriate to compare just the mean grey value to see an endocytic activity. The mean grey value could be higher in larger cells with a larger area of the PM even if the endocytic rate is constant between these cell types.

We thank you for this comment. We agree that an increase in endocytic membrane flow to the vacuole may not be dependent on an increase in endocytosis rate. We improved the discussion on this matter in the revised version of the manuscript. When compared to the cell size increase, we however observed accelerated size increase of BFA compartments during cellular enlargements, rather suggesting that a reprogramming of intracellular membrane trafficking contributes to this effect. Please, also see additional comments in reply to reviewer 1 and 2.